# GuardHFL: Privacy Guardian for Heterogeneous Federated Learning

## Abstract

Heterogeneous federated learning (HFL) enables clients with different computation and communication capabilities to collaboratively train their own customized models via a query-response paradigm on auxiliary datasets. However, such paradigm raises serious privacy issues due to the leakage of highly sensitive query samples and response predictions. Although existing secure querying solutions may be extended to enhance the privacy of HFL with non-trivial adaptation, they suffer from two key limitations: (1) lacking customized protocol designs and (2) relying on heavy cryptographic primitives, which could lead to poor performance. In this work, we put forth GuardHFL, the *first-of-its-kind* efficient and privacy-preserving HFL framework. GuardHFL is equipped with a novel HFL-friendly secure querying scheme that is built on lightweight secret sharing and symmetric-key techniques. Its core is a set of customized multiplication and comparison protocols, which substantially boost the execution efficiency. Extensive evaluations demonstrate that GuardHFL outperforms the state-of-the-art works in both runtime and communication overhead.

## 1 Introduction

As a promising variant of federated learning (FL), heterogeneous federated learning (HFL) (Li & Wang, 2019) enables clients equipped with different computation and communication capabilities to collaboratively train their own customized models that may *differ in size, numerical precision or structure* (Lin et al., 2020). In particular, the knowledge of models is shared via a query-response paradigm on auxiliary datasets, such as unlabeled datasets from the same task domain (Choquette-Choo et al., 2021) or related datasets from different task domains (Li & Wang, 2019; Lin et al., 2020). In such a paradigm, each client queries others with samples in the auxiliary querying dataset, and obtains aggregated response predictions via a centralized cloud server[1]. Then he retrains his local model on the query data and corresponding predictions. This flexible approach facilitates customized FL-driven services in areas like healthcare and finance (Kairouz et al., 2019), while resolving the intellectual property concerns of FL models (Tekgul et al., 2021).

However, HFL suffers from several privacy issues. First, directly sharing query samples violates their privacy. For example, in healthcare applications, the auxiliary datasets may contain patients' medical conditions. Disclosure of such information is illegal under current regulations like General Data Protection Regulation. Second, sharing response predictions may still compromise the privacy of local data (Papernot et al., 2016). Several works have shown that given black-box access to a model, adversaries can infer the membership (Salem et al., 2019) and attribute information (Ganju et al., 2018) of the target sample or even reconstruct the original training data (Yang et al., 2019).

Although in traditional FL systems, the privacy issue could be mitigated through well-studied secure gradient aggregation protocols (Bell et al., 2020), it becomes more challenging to realize this guarantee in HFL, due to the heterogeneity of the clients' models (refer to Appendix A.2.3). To bridge this gap, a possible solution is to structurally integrate into HFL existing secure querying (a.k.a. private inference) schemes (Rathee et al., 2020; Huang et al., 2022; Wagh et al., 2019; Tan et al., 2021). These schemes utilize various cryptographic primitives, including homomorphic encryption

---

[1]As demonstrated in Bonawitz et al. (2017); Bell et al. (2020), clients (e.g., mobile devices) in real-world applications are generally widely distributed and coordinated only by the server.

(HE) (Gentry, 2009), garbled circuit (GC) (Yao, 1986) or oblivious transfer (OT) (Asharov et al., 2013), to provide rigorous privacy guarantees for query data and prediction results. While it is possible to non-trivially extend these secure querying schemes (refer to Section 2.3), they have two major limitations: (1) the underlying protocols are not customized for HFL; (2) they incur huge overhead due to the adoption of heavy cryptographic primitives. These bottlenecks lead to poor performance and hinder the efficient instantiation of HFL. Therefore, it is necessary but challenging to provide customized protocols and implement a privacy-preserving HFL with desirable performance.

In this work, we introduce `GuardHFL`, the *first* efficient and privacy-preserving HFL framework to address the above challenges[2]. `GuardHFL` is built upon the standard HFL training paradigm (Li & Wang, 2019), which contains three stages: local training, querying and local re-training (refer to Section 2.1). We formalize the workflow of HFL, and present a novel HFL-friendly secure querying scheme as an important building block. The core of our scheme is a set of customized multiplication and comparison protocols, which substantially boost the execution efficiency compared to existing works. More precisely, we optimize the parallel prefix adder (PPA) logic (Harris, 2003) to build a more advanced comparison protocol. Over an $\ell$-bit ring, these optimizations reduce the evaluation of $\log \ell$ AND gates and also the number of the communication rounds, which are two crucial factors that dominate the performance. Moreover, our PRF-based multiplication protocol only communicates 3 elements in an $\ell$-bit ring, achieving significant improvement compared to the widely used Beaver's multiplication solution (Beaver, 1991; Demmler et al., 2015). We provide formal privacy guarantees for the designed protocols, and evaluate `GuardHFL` on different datasets (SVHN, CIFAR10, Tiny ImageNet), system configurations (IID and Non-IID training sets) and heterogeneous models. Results show that `GuardHFL` outperforms the state-of-the-art works in efficiency while ensuring the model utility.

## 2 BACKGROUND

### 2.1 HETEROGENEOUS FEDERATED LEARNING

We briefly review the workflow of HFL (Li & Wang, 2019), where clients independently design their own unique models. Due to such model heterogeneity, clients cannot directly share model parameters with each other as in the traditional FL. Instead, they learn the knowledge of other models via a query-response mechanism, which is similar as the knowledge distillation technique (Hinton et al., 2015). To be more precise, each client $P_Q$ (called the querying party) performs three-phase operations collaboratively with a server: (1) *Local training*: $P_Q$ first trains the local model on his private dataset. (2) *Querying*: The server selects $C$ fraction of clients as the responding parties $P_A$. Given the auxiliary querying dataset, the server receives the prediction results from these $P_A$, computes the aggregated result and returns it back to $P_Q$. (3) *Local re-training*: $P_Q$ then retrains the local model based on the private dataset, as well as the query samples and corresponding predictions. Each client in HFL can play the roles of the querying party and the responding party at the same time, and the above process is iterated until each local model meets the pre-defined accuracy requirement. Note that as illustrated in existing works (Bonawitz et al., 2017; Bell et al., 2020), the server is responsible for routing the messages between clients, since the clients (e.g., mobile devices) generally cannot establish direct communication channels with others.

`GuardHFL` is in line with the above paradigm with the additional benefit of privacy protection. The only difference lies in the acquisition of auxiliary query samples in the querying stage. In general HFL (Li & Wang, 2019), there is a large public auxiliary dataset (used as query samples) that every party can access. However, considering the privacy limitation, such public dataset is hard to collect in real-world scenarios such as healthcare. To tackle this problem, in `GuardHFL`, each party can locally construct a synthesized querying set based on its private training samples, by utilizing existing data augmentation strategies (refer to Section 3.4).

### 2.2 THREAT MODEL

As described in Section 1, in the querying phase of HFL, the query samples, prediction results and model parameters may contain sensitive information that is of interest to adversaries. In line

---

[2]Choquette-Choo et al. (2021) presented a general collaborative learning scheme, called CaPC, which enables each party to improve his local model from others' models using the secure querying scheme (Boemer et al., 2019b). However, it cannot be directly applied to the HFL scenario as it requires cross-client communication. Meanwhile, it causes prohibitively high overhead (refer to Section 4.1).

Figure 1: High-level view of `GuardHFL`

with prior works (Phong et al., 2018; Sun & Lyu, 2021; Choquette-Choo et al., 2021), we consider an honest-but-curious adversary setting (Goldreich, 2009), where each entity (including the clients and the server) strictly follows the specification of the designed protocol but attempts to infer more knowledge about these private information of other clients. Moreover, to maintain the reputation and provide more services, the server does not collude with any clients, namely that an attacker either corrupts the server or a subset of clients but not both.

## 2.3 EXTENDING EXISTING SECURE QUERYING SOLUTIONS TO HFL

To provide privacy guarantees against adversaries in Section 2.2, the clients and the server need to privately execute the querying process. Although this process consists of three entities (i.e., $P_Q$, the server and $P_A$), it is non-trivial to directly extend existing secure 3-party computation protocols (3PC) (Wagh et al., 2019; 2021; Knott et al., 2021; Tan et al., 2021) to instantiate this process. The main reason is the incapability of direct communication between $P_Q$ and $P_A$ in realistic HFL scenarios (Bonawitz et al., 2017; Bell et al., 2020), which hinders the usage of these 3PC solutions in HFL, unless we redesign the underlying protocols and make substantial modifications to their corresponding implementations. On the other hand, we can extend state-of-the-art 2PC solutions (Rathee et al., 2020; Huang et al., 2022) into this process via using the server as the communication medium with adaptive protocol modifications (refer to Appendix A.2.4 for more details). Unfortunately, as mentioned in Section 1, such extension comes at the cost of heavy computational and communication complexity. Motivated by these challenges, we design a set of lightweight and customized protocols for improving the efficiency of the secure querying phase (Section 3), which show significant performance gains over extending the advanced 2PC schemes to HFL (Section 4.1).

## 2.4 CRYPTOGRAPHIC PRIMITIVES

**Secret sharing.** `GuardHFL` adopts the 2-out-of-2 arithmetic secret sharing scheme over the ring $\mathbb{Z}_{2^\ell}$ (Shamir, 1979; Demmler et al., 2015). $\text{Share}(x)$ denotes the sharing algorithm that takes $x$ as input and outputs random sampled shares $[x]_0, [x]_1$ with the constraint $x = [x]_0 + [x]_1$ in $\mathbb{Z}_{2^\ell}$. The reconstruction algorithm $\text{Recon}([x]_0, [x]_1)$ takes the two shares as input and outputs $x = [x]_0 + [x]_1$ in $\mathbb{Z}_{2^\ell}$. Besides, our comparison protocol adopts the boolean secret sharing (Shamir, 1979; Demmler et al., 2015), where $x \in \mathbb{Z}_2$ is shared as $[x]_0^B$ and $[x]_1^B$ satisfying $[x]_0^B \oplus [x]_1^B = x$ in $\mathbb{Z}_2$. The security ensures that given $[x]_0$ or $[x]_1$ (similarly, $[x]_0^B$ or $[x]_1^B$), the value of $x$ is perfectly hidden. Arithmetic operations on secret-shared values can be implemented with existing techniques (Appendix A.2.2).

**Pseudo-random Function**. A pseudo-random function $y \leftarrow \text{PRF}(Sk, x)$ is a deterministic function that takes a uniformly random seed $Sk$ and a payload $x$ as input and outputs a fixed-length pseudo-random string $y$. The security of PRFs ensures that the output is indistinguishable from the uniform distribution. In `GuardHFL`, PRFs enable two parties to generate the same pseudo-random values without communication. Details can be found in Appendix A.2.2.

## 3 GUARDHFL

`GuardHFL` is built upon standard HFL systems as shown in Section 2.1 and enhances their privacy protections with cryptographic techniques. Figure 1 shows the overview of `GuardHFL` and the detailed description is given in Algorithm 1. Similar as vanilla HFL, it includes three phases: *Local training*, *Secure querying* and *Local re-training*. Since *Local training* and *Local re-training* are standard HFL training processes without privacy issues, below we focus on formalizing our core construction, i.e., *Secure querying*. As detailed in Section 2.3, extending existing secure querying solutions to HFL introduces expensive overhead due to the usage of heavy cryptographic primitives and the lack of customized protocols. To tackle this challenge, we propose a tailored secure querying scheme utilizing lightweight secret sharing and PRF techniques, which is decomposed into three steps: *secure query-data sharing*, *secure model prediction* and *secure result aggregation*.

In generally, $P_Q$ first constructs querying samples locally using data argumentation strategies (Section 3.4). Since querying samples imply the semantic information of private training data, they

---

**Algorithm 1** The GuardHFL framework

---

**Input:** Each client $P_j$, $j \in [n]$, holds a private dataset $\mathcal{D}_j$ and a customized local model $M_j$. *iter* is the number of iterations. $B$ is the number of query samples and $\mathcal{C}$ is the set of selected responding parties in the current query-response phase.

**Output:** Trained models $M_j$, $j \in [n]$.

1: **for** each $j \in [n]$ **do**
2:    $P_j$ locally trains the local model $M_j$ on $\mathcal{D}_j$ using the stochastic gradient descent optimization.
3: **end for**
4: **for** each *iter* **do**
5:    **for** each querying party $P_Q^j$, $j \in [n]$ **do**
6:       $P_Q^j$ randomly samples query data $\{x_b\}_{b \in [B]}$ from the auxiliary querying dataset that are generated via the data argumentation strategies described in Section 3.4.
7:       **for** each responding party $P_A^i$, $i \in \mathcal{C}$ **do**
8:          $P_Q^j$ secret-shares $\{[x_b]\}_{b \in [B]}$ with $P_A^i$ and the server, based on protocol $\Pi_{\text{Share}}$ in Figure 2.
9:          $P_A^i$, $P_Q^j$ and the server jointly perform the secure model prediction protocol in Section 3.2.
10:          $P_A^i$ secret-shares the predictions $\{[y_b^i]\}_{b \in [B]}$ to $P_Q$ and the server.
11:       **end for**
12:       $P_Q^j$ computes $\{y_b\}_{b \in [B]}$, where $y_b = \sum_{i \in \mathcal{C}} y_b^i$, via protocol $\Pi_{\text{Agg}}$ in Figure 5 with the server.
13:       $P_Q^j$ retrains $M_j$ based on the query dataset $\{x_b, y_b\}_{b \in [B]}$ and $\mathcal{D}_j$.
14:    **end for**
15: **end for**

---

cannot be directly exposed to the server and $P_A$ for prediction. Therefore, GuardHFL secret-shares the query samples to the server and $P_A$ using the designed secure query-data sharing protocol (Section 3.1). Then given the secret-shared samples, $P_A$, $P_Q$ and the server can jointly execute the proposed secure model prediction protocol (Section 3.2) to obtain the secret-shared inference logits. After that, the secure result aggregation protocol (Section 3.3) comes in handy, which takes as input the secret-shared logits and returns the aggregated results to $P_Q$. During the entire querying phase, GuardHFL maintains the following *invariant* (Rathee et al., 2020; Huang et al., 2022): the server and $P_A$ start each protocol with arithmetic shares of inputs, and end with arithmetic shares of outputs over the same ring. This allows us to sequentially stitch the proposed protocols to obtain a fully private querying scheme. The formal security analysis is given in Appendix A.3.

### 3.1 SECURE QUERY-DATA SHARING

To perform secure prediction based on secret sharing techniques, $P_Q$ first secret-shares the query data $x$ with the server and $P_A$. Considering the communication constraint between $P_Q$ and $P_A$, we utilize PRFs to share $x$. Specifically, we first construct PRF seeds in pairs for $P_Q$, $P_A$ and the server, denoted as $Sk_{QA}$, $Sk_{SA}$, and $Sk_{SQ}$, which are used to generate the same random values between two parties without communication (refer to Figure 12 in Appendix A.2.2). After that, $P_Q$ can share $x$ using protocol $\Pi_{\text{Share}}$ as shown in Figure 2. In particular, $P_Q$ non-interactively shares $[x]_0 = r$ with $P_A$ using PRFs on the seed $Sk_{QA}$. Then $P_Q$ computes $[x]_1 = x - r$ and sends it to the server.

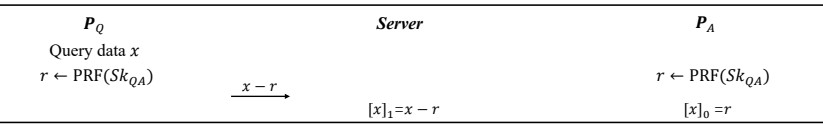

Figure 2: Secure query-data sharing protocol $\Pi_{\text{Share}}$

### 3.2 SECURE MODEL PREDICTION

In this step, the server and $P_A$ execute secure prediction on the secret-shared query data with the assistance of $P_Q$. Figure 11 in Appendix A.2.1 gives an end-to-end graphic depiction. Below we elaborate the customized protocols for three components: linear layers, ReLU and MaxPooling.

**Linear layers.** Linear layers consist of fully-connection, convolution, average pooling and batch normalization, and the main operation of these layers is matrix multiplication (Wagh et al., 2019; Huang et al., 2022). We utilize PRFs and secret sharing to design a customized matrix multiplication protocol $\Pi_{\text{Matmul}}$, which is not only compatible with communication-limited HFL settings, but also improves communication efficiency. Specifically, as shown in Figure 3, $P_A$ and the server aim to

compute $\omega x$, where the model parameter $\omega$ is held by $P_A$ and the shares $[x]_0, [x]_1$ of $x$ are held by $P_A$ and the server, respectively. Given that $\omega x = \omega[x]_0 + \omega[x]_1$, $P_A$ can compute $\omega[x]_0$ locally. To evaluate $\omega[x]_1$, $P_Q$ first generates three random matrices $a$, $b$ and $[c]_0$ using PRFs, meanwhile, computes and sends $[c]_1$ that satisfies $[c]_1 + [c]_0 = ab$ in $\mathbb{Z}_{2^\ell}{}^3$ to the server. At the same time, using PRFs, the server generates the same $b$ and $P_A$ generates the same $a$ and $[c]_0$. Then $P_A$ and the server can learn $[y]_0$ and $[y]_1$ (i.e., the secret shares of $\omega x$), respectively, through one round of interaction. Overall, the communication cost is $3\ell$ within 1 communication round.

*Remark.* In our fixed-point representations, to prevent values from overflowing due to the multiplication operations, we use the truncation technique from Mohassel & Zhang (2017), which is consistent with existing methods (Mishra et al., 2020; Wagh et al., 2019). This technique simply truncates the extra LSBs of fixed-point values, albeit at the cost of a 1-bit error, which is negligible on the model accuracy (Mohassel & Zhang, 2017).

| $P_Q$ | Server | $P_A$ |
|---|---|---|
| | $[x]_1$ | Model parameter $\omega$, $[x]_0$ |
| $a, [c]_0 \leftarrow \mathrm{PRF}(Sk_{QA})$ | | $a, [c]_0 \leftarrow \mathrm{PRF}(Sk_{QA})$ |
| $b \leftarrow \mathrm{PRF}(Sk_{SQ})$ | $b \leftarrow \mathrm{PRF}(Sk_{SQ})$ | |
| $[c]_1 = ab - [c]_0 \longrightarrow$ | $\overset{\omega+a}{\longleftarrow}$ $\overset{[x]_1-b}{\longrightarrow}$ | |
| | $[y]_1 = (\omega+a)b - [c]_1$ | $[y]_0 = \omega[x]_0 + \omega([x]_1 - b) - [c]_0$ |

Figure 3: Secure matrix multiplication protocol $\Pi_{\text{Matmul}}$

**ReLU.** ReLU is computed as $\mathsf{ReLU}(x) = x \cdot (1 \oplus \mathsf{MSB}(x))$, where $\mathsf{MSB}(x)$ equals 0 if $x \geq 0$ and 1 otherwise. Thus, the evaluation of ReLU consists of MSB and multiplication operations. In the following, we propose an optimized MSB protocol building on the parallel prefix adder (PPA) logic (Harris, 2003; Mohassel & Rindal, 2018; Patra et al., 2021). Before giving specific optimizations, we first review the PPA-based MSB method.

*PPA-based MSB method.* Given that the bit decomposition of $[x]_0$ and $[x]_1$ are $e_\ell, \ldots, e_1$ and $f_\ell, \ldots, f_1$, respectively, the PPA-based method evaluates $\mathsf{MSB}(x) = e_\ell \oplus f_\ell \oplus c_\ell$, where $c_\ell$ is the $\ell$-th carry bit. Note that $c_\ell = c_{\ell-1} \wedge (e_{\ell-1} \oplus f_{\ell-1}) \oplus (e_{\ell-1} \wedge f_{\ell-1})$. Given the *carry signal tuple* $(g_i^0, p_i^0)$ where $g_i^0 = e_i \wedge f_i$ and $p_i^0 = e_i \oplus f_i$ for $i \in [\ell]$, $c_\ell$ can be reformulated as

$$c_\ell = g_{\ell-1}^0 \oplus (p_{\ell-1}^0 \wedge g_{\ell-2}^0) \oplus \cdots \oplus (p_{\ell-1}^0 \wedge \cdots \wedge p_2^0 \wedge g_1^0). \tag{1}$$

The PPA evaluates Eq.1 by constructing a $\log \ell$-layer tree and traversing from the leaves with inputs $(g_i^0, p_i^0)$ for $i \in [\ell]$ until reaching the root. Each node $k$ at the $n$-th layer ($n \in [\log \ell]$) is adhered with the following operation:

$$g_k^n = g_{j+1}^{n-1} \oplus (g_j^{n-1} \wedge p_{j+1}^{n-1}) \text{ and } p_k^n = p_{j+1}^{n-1} \wedge p_j^{n-1}, \tag{2}$$

where it takes as inputs of two adjacent signal tuples $(g_{j+1}^{n-1}, p_{j+1}^{n-1})$ and $(g_j^{n-1}, p_j^{n-1})$, and outputs a signal tuple $(g_k^n, p_k^n)$. Finally, $c_\ell$ is obtained as $g_1^{\log \ell}$, and $\mathsf{MSB}(x) = e_\ell \oplus f_\ell \oplus g_1^{\log \ell}$. Overall, this method requires $3\ell - 4$ AND gates within $\log \ell + 1$ communication rounds, where each AND gate is evaluated by the standard Beaver triple-based multiplication protocol (Appendix A.2.2). Therefore, it totally needs $30\ell - 40$ bits communication.

*Optimizations.* Straightforwardly adopting the above PPA-based MSB method cannot achieve the best efficiency. The performance bottleneck comes from: (1) unnecessary AND gates evaluation introduces extra communication overhead; (2) more communication rounds due to the separation of input computation and tree evaluation. To solve these challenges, firstly, we simplify the PPA circuit via removing unnecessary AND gates. In each level $n \in \{1, 2, \cdots, \log \ell\}$, we eliminate the generation of "least significant" $p_1^n$ that consumes an evaluation of AND but it is unnecessary for computing MSB. For the same reason, we also remove the evaluation of $g_\ell^0$. Secondly, we can further reduce the communication complexity by utilizing the above multiplication protocol $\Pi_{\text{Matmul}}$ to evaluate remaining AND gates, instead of the Beaver triple-based method. Thirdly, we integrate the round of communication used to compute inputs $g_i^0$ for $i \in \{1, 2, \cdots, \ell-1\}$ into the evaluation of the PPA circuit via a non-trivial modification of the evaluation logic. Overall, our MSB protocol totally communicates $9\ell - 3\log \ell - 12$ bits within $\log \ell$ rounds, a $3.4\times$ communication improvement over the above PPA-based MSB method. Algorithm 2 gives the detailed construction of $\Pi_{\text{msb}}$.

---

[3] $(a, b, [c]_0, [c]_1)$ with the constrain $c = ab$ in $\mathbb{Z}_{2^\ell}$ can be seen as a variant of the Beaver's multiplication triple. Details refer to Appendix A.2.2.

---

**Algorithm 2** Secure MSB Protocol $\Pi_{\text{msb}}$

---

**Input:** The arithmetic shares $[x]$
**Output:** The boolean shares $[\text{msb}(x)]^B$
1: Initiate $g^*$ and $p^*$ with size $\ell$. Let $g_i^*$ and $p_i^*$ are the $i$-th positions of $g^*$ and $p^*$ respectively.
2: $P_0$ and $P_1$ set the bit decomposition of $[x]_0$ and $[x]_1$ to $e_\ell, \cdots, e_1$ and $f_\ell, \cdots, f_1$ respectively.
3: $P_0$ and $P_1$ invoke $\Pi_{\text{AND}}$ with inputs $e_i$ and $f_i$ to obtain $[g_i^*]^B$ for $i \in [\ell - 1]$.
4: $P_0$ sets $[p_i^*]_0^B = e_i$ and $P_1$ sets $[p_i^*]_1^B = f_i$.
5: $P_0$ and $P_1$ invoke $\Pi_{\text{AND}}$ with inputs $[p_{2i-1}^*]^B$ and $[p_{2i-2}^*]^B$ to obtain $[p_i^*]^B$ for $i \in \{2, \cdots, \frac{\ell}{2}\}$.
6: **for** $r \in \{2, \cdots, \log \ell\}$ **do**
7:     **for** $i \in \{2, \cdots, \frac{\ell}{2^{r-1}}\}$ **do**
8:         **If** $r = 2$, $P_0$ and $P_1$ invoke $\Pi_{\text{AND}}$ with inputs $[g_{2i-2}^*]^B$ and $[p_{2i-1}^*]^B$ to obtain $[t_i]^B$, and sets $[g_i^*]^B = [g_{2i-1}^*]^B \oplus [t_i]^B$. **Else** $P_0$ and $P_1$ invoke $\Pi_{\text{AND}}$ with inputs $[g_{2i-1}^*]^B$ and $[p_{2i}^*]^B$ to obtain $[t_i]^B$, and sets $[g_i^*]^B = [g_{2i}^*]^B \oplus [t_i]^B$.
9:     **end for**
10:     $P_0$ and $P_1$ invoke $\Pi_{\text{AND}}$ with inputs $[g_1^*]^B$ and $[p_2^*]^B$ to obtain $[t_1]^B$, and sets $[g_1^*]^B = [g_2^*]^B \oplus [t_1]^B$.
11:     **for** $i \in \{2, \cdots, \frac{\ell}{2^{r-1}}\}$ **do**
12:         **If** $r = 2$, $P_0$ and $P_1$ invoke $\Pi_{\text{AND}}$ with inputs $[p_{2i-1}^*]^B$ and $[p_{2i-2}^*]^B$ to obtain $[p_i^*]^B$. **Else** $P_0$ and $P_1$ invoke $\Pi_{\text{AND}}$ with inputs $[p_{2i}^*]^B$ and $[p_{2i-1}^*]^B$ to obtain $[p_i^*]^B$.
13:     **end for**
14: **end for**
15: $P_0$ sets $[\text{msb}(x)]_0^B = e_\ell \oplus [g_1^*]_0^B$ and $P_1$ sets $[\text{msb}(x)]_1^B = f_\ell \oplus [g_1^*]_1^B$.

---

After obtaining $[\text{MSB}(x)]^B$, we need to compute $[x] \cdot (1 \oplus [\text{MSB}(x)]^B)$, i.e., the secret shares of ReLU$(x)$. Given $z_0 = [\text{MSB}(x)]_0^B$ and $z_1 = 1 \oplus [\text{MSB}(x)]_0^B$ for simplicity, we have ReLU$(x) = ([x]_0 + [x]_1)(z_0 + z_1 - 2z_0 z_1) = z_0[x]_0 + z_1[x]_1 + z_1(1 - 2z_0)[x]_0 + z_0(1 - 2z_1)[x]_1$. The first two terms can be evaluated locally by $P_A$ and the server respectively, while the other two terms are evaluated using protocol $\Pi_{\text{Matmul}}$. For example, to compute $z_1(1 - 2z_0)[x]_0$, the protocol $\Pi_{\text{Matmul}}$ inputs $t_0 = (1 - 2z_0)[x]_0$ and $z_1$ provided by $P_A$ and the server, and outputs $[t_0 z_1]_0$ and $[t_0 z_1]_1$ to $P_A$ and the server, respectively. Finally, $P_A$ and the server learn $[y]_0$ and $[y]_1$ respectively, such that $y = \text{ReLU}(x)$. The complete secure ReLU protocol $\Pi_{\text{ReLU}}$ is shown in Figure 4.

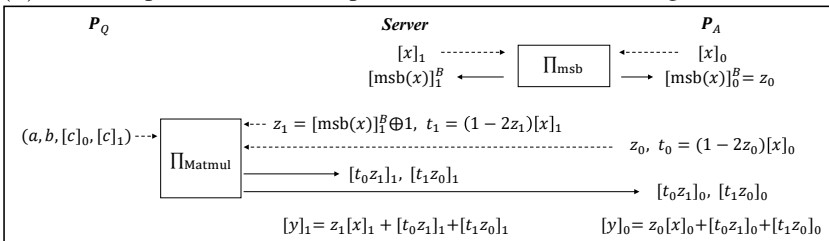

Figure 4: Secure ReLU protocol $\Pi_{\text{ReLU}}$

**Maxpooling.** The evaluation of Maxpooling can be performed with the protocol $\Pi_{\text{ReLU}}$ as well as a tree-based round optimization that recursively partitions the values into two halves and then compares the elements of each half. Precisely, the entities arrange the input of $m$ elements into a 2-ary tree with the depth of $\log m$, and evaluate the tree in a top-down fashion. In each comparison of two secret-shared elements $[x]$ and $[y]$, we utilize the observation of $\max([x], [y]) = \text{ReLU}([x] - [y]) + [y]$. Hence the protocol complexity of Maxpooling mainly comes from the evaluation of $m - 1$ ReLU. Besides, as illustrated in Wagh et al. (2019); Mishra et al. (2020), AvgPooling can be evaluated locally without communication.

### 3.3 SECURE RESULT AGGREGATION

After the secure prediction, the predicted logit $[x_i]$ is secret-shared between the server and each responding party $P_A^i$, where $i \in [C]$ and $C$ is the set of responding parties. To prevent privacy leakage from a single prediction (Salem et al., 2019; Ganju et al., 2018; Yang et al., 2019), we return the aggregated logit to $P_Q$ via the secure aggregation protocol $\Pi_{\text{Agg}}$ in Figure 5. Specifically, $P_A^i$ and $P_Q$ first generate a random value $r_i$ based on PRFs. Then each $P_A^i$ computes and sends $[x_i]_0 - r_i$ to the server. The server sums all received values and sends the masked aggregation to $P_Q$, which will reconstruct the aggregated logits of the query data. Notice that our secure aggregation protocol can be extended to output the aggregated label rather than the logit, using the above $\Pi_{\text{ReLU}}$ protocol.

$$P_Q \qquad\qquad Server \qquad\qquad P_A^i$$

$$[x_i]_1, i \in [C] \qquad\qquad [x_i]_0$$

$$r_i \leftarrow \text{PRF}(Sk_{QA}) \qquad\qquad r_i \leftarrow \text{PRF}(Sk_{QA})$$

$$\xleftarrow{\quad [x_i]_0 - r_i \quad}$$

$$\xleftarrow{\quad k \quad} \quad k = \sum_{i=1}^{|C|}([x_i]_0 - r_i + [x_i]_1)$$

$$y \leftarrow \text{softmax}(k + \sum_{i=1}^{|C|} r_i)$$

Figure 5: Secure result aggregation protocol $\Pi_{\text{Agg}}$

## 3.4 DISCUSSION

**Query data construction.** Unlike existing HFL works relying on auxiliary datasets as the query data (Choquette-Choo et al., 2021; Lin et al., 2020), we demonstrate the feasibility of model knowledge transfer in `GuardHFL` by constructing a synthesized query set based on private training data, to alleviate potential limitations (e.g., privacy, acquisition and storage) of auxiliary datasets. A simple solution is to directly use the private training data to query, like well-studied knowledge distillation (Hinton et al., 2015). Moreover, we also construct a synthesized dataset via the mixup method (Zhang et al., 2018) (Appendix A.1.2). The synthesized dataset construction is a universal and modular method, and it can be readily extended with advanced data augmentation strategies, such as cutout (DeVries & Taylor, 2017) and cutmix (Yun et al., 2019). Note that this process does not reveal any private information, since the samples are constructed locally by the querying party, without involving any other parties and their private datasets. We present some exploration and experiments in Appendix A.1.2 and Figure 10(c).

**GPU-friendly evaluation.** Our scheme is friendly with GPUs and can be processed by highly-optimized CUDA kernels (Tan et al., 2021). As discussed above, the cryptographic protocols of `GuardHFL` only involve simple vectorized arithmetic operations, rather than homomorphic encryption and garbled circuits in prior works (Rathee et al., 2020; Huang et al., 2022; Choquette-Choo et al., 2021). As a result, `GuardHFL` is suitable for batch querying (i.e., executing multiple querying at the same time) with a lower amortized cost. We evaluate the designed protocols on GPUs in Section 4.1 and show the advantage of GPU acceleration over CPUs in Appendix A.1.2.

## 4 EVALUATION

**Datasets and models.** We evaluate `GuardHFL` on three image datasets (SVHN, CIFAR10 and Tiny ImageNet). By default, we assume independent and identically distributed (IID) training data among clients. We also simulate disjoint Non-IID training data via the Dirichlet distribution $\text{Dir}(\alpha)$ in Lin et al. (2020). The value of $\alpha$ controls the degree of Non-IID-ness, where a smaller $\alpha$ indicates a higher degree of Non-IID-ness. Moreover, we simulate the heterogeneity property in HFL. In particular, for SVHN and CIFAR10, we set the number of clients $n = 50$ and use VGG-7, ResNet-8 and ResNet-10 architectures as the clients' local models. For Tiny ImageNet, we set $n = 10$ and use ResNet-14, ResNet-16, and ResNet-18 architectures. Each model architecture is used by $n/3$ clients. Besides, the query data are constructed via two methods as shown in Section 3.4: using the private training data (Q-priv) or synthesizing samples (Q-syn) via mixup (Zhang et al., 2018).

**Cryptographic protocol.** Following existing works (Rathee et al., 2020; Tan et al., 2021), we set secret-sharing protocols over a 64-bit ring $\mathbb{Z}_{2^{64}}$, and encode inputs using a fixed-point representation with 20-bit precision. The security parameter $\kappa$ is 128 in the instantiation of PRFs. Unless otherwise stated, we only report the performance on the GPU accelerator. More experimental setup is given in Appendix A.1.1.

## 4.1 EFFICIENCY

We report the efficiency of `GuardHFL`, and compare it with CaPC (Choquette-Choo et al., 2021) and instantiations of HFL based on state-of-the-art secure querying protocols (Rathee et al., 2020; Huang et al., 2022; Tan et al., 2021).

**End-to-end performance.** We show the extra overhead introduced by `GuardHFL` compared with the vanilla HFL system in the plaintext environment. This is caused by the secure querying phase, which consists of three steps described in Section 3. Table 1 reports the runtime of each step for different models and datasets[4]. We observe that the cost is dominated by the secure model prediction step. Specifically, it takes 16.9 minutes to evaluate 5000 query samples securely on VGG-7 and CIFAR10, and only 11.32 second and 0.3 second are spent on the secure query-data sharing and secure result aggregation steps. More time is required to evaluate Tiny ImageNet because of larger input sizes and model architectures.

---

[4]To clearly illustrate the efficiency of `GuardHFL`, unless otherwise specified, we only show the overhead of one user in each iteration as described in Section 3.

Table 1: Extra runtime (sec) of `GuardHFL` over vanilla HFL systems in the plaintext environment. CIFAR10 and SVHN have the same runtime due to the same input size and model architecture.

| Dataset | # of Queries | 1. Query data sharing | 2. Secure prediction | | | 3. Result aggreg. |
|---|---|---|---|---|---|---|
| | | | VGG-7 | RESNET-8 | RESNET-10 | |
| CIFAR10 (SVHN) | 1000 | 5.08 | 205.46 | 270.78 | 305.46 | 0.09 |
| | 2500 | 7.16 | 511.63 | 657.83 | 758.16 | 0.12 |
| | 5000 | 11.32 | 1019.12 | 1346.79 | 1521.23 | 0.30 |
| | | | RESNET-14 | RESNET-16 | RESNET-18 | |
| TINY IMAGENET | 1000 | 9.87 | 2700.96 | 2971.47 | 3084.81 | 0.18 |
| | 2500 | 18.78 | 6815.69 | 7217.28 | 7503.5 | 0.32 |

Table 2: Comparison with CaPC on runtime (sec) over MNIST and three heterogeneous models as the batch size (BS) of query data increases.

| Model | CryptoNets | | CryptoNets-ReLU | | MLP | |
|---|---|---|---|---|---|---|
| | GUARDHFL | CAPC | GUARDHFL | CAPC | GUARDHFL | CAPC |
| BS=128 | **0.03** | 17.75 | **0.24** | 48.83 | **0.75** | 65.01 |
| BS=256 | **0.05** | 17.56 | **0.31** | 70.14 | **0.83** | 86.37 |
| BS=512 | **0.07** | 17.62 | **0.50** | 112.42 | **1.05** | 129.81 |
| BS=1024 | **0.13** | 17.77 | **0.89** | 201.42 | **1.58** | 216.61 |

**Comparison with CaPC.** As described in Section 1, similar to `GuardHFL`, CaPC (Choquette-Choo et al., 2021) was proposed to support private collaborative learning utilizing the secure querying scheme (Boemer et al., 2019b), but with the unrealistic cross-client communication. In Table 2, we compare the secure querying process of `GuardHFL` with CaPC. Following CaPC's setup, we evaluate three small-scale models (CryptoNets (Gilad-Bachrach et al., 2016), CryptoNets-ReLU (Gilad-Bachrach et al., 2016) and MLP (Boemer et al., 2019b)) on MNIST. We observe that `GuardHFL` is two orders of magnitude faster than CaPC on the three models. In terms of communication overhead, we provide a theoretical comparison. (1) For linear layers, CaPC requires to communicate 2 homomorphic ciphertexts within 2 rounds. `GuardHFL` needs communicating 3 ring elements (each with 64-bit). Note that the size of ciphertexts is much larger than the size of the ring elements. (2) For non-linear layers, e.g., ReLU, CaPC adopts the garbled circuit technique that requires 2 rounds with $8\ell\lambda - 4\lambda$ communication bits ($\lambda = 128$ and $\ell = 64$ in our setting) (Rathee et al., 2020). `GuardHFL` only requires communicating $15\ell - 3\log\ell - 12$ bits, a $70\times$ improvement over CaPC.

**Comparison with SOTA works.** To further demonstrate the efficiency of `GuardHFL`, we instantiate HFL based on SOTA 2PC schemes, including Cheetah (Huang et al., 2022) and CrypTFlow2 (Rathee et al., 2020), using the methods described in Appendix A.2.4. Table 3 reports the runtime and communication comparison of the secure prediction phase over CIFAR10. We observe that `GuardHFL` achieves a significant efficiency improvement on three heterogeneous models. For example, `GuardHFL` requires 57.4∼75.6× less runtime and 8.6∼12.7× less communication compared to CrypTFlow2. This is because the latter needs heavy HE-based multiplication and OT-based comparison within multi-communication rounds. Moreover, as shown in Section 2.3, extending 3PC protocols such as CryptGPU (Tan et al., 2021) to HFL is non-trivial. However, since GryptGPU is one of the most advanced protocols under GPU analogs built on CrypTen (Knott et al., 2021), we also compare with it assuming no communication limitation. We would like to mention that despite such an unfair comparison, `GuardHFL` still has performance advantages, i.e., roughly 2.1× and 2.0× in computation and communication overheads, respectively.

## 4.2 ACCURACY

We report the accuracy of each heterogeneous model in `GuardHFL`, and explore the impact of various factors on the model accuracy such as the Non-IID setting, and the number of query data.

**End-to-end model accuracy.** Table 4 reports the model accuracy on three datasets in `GuardHFL`. We observe that for SVHN and CIFAR10, using Q-priv to query can increase the accuracy by about 4%, while the accuracy gain is about 10% when using 10K query samples with Q-syn. The main reason is that synthetic samples could provide a good coverage of the manifold of natural data. We also observe that more synthetic query data can achieve better performance from Table 4. Furthermore, with an increased number of participating clients, the accuracy improves slightly. Figure 6 shows the accuracy curves versus the number of iterations. We use SVHN and CIFAR10 as examples, as they converge much faster with better readable curves than Tiny ImageNet. We can observe that each heterogeneous model on both datasets can converge well based on two types of query data, and Q-syn shows better performance.

Table 3: Comparison with advanced secure prediction protocols on runtime (sec) and communication (MB) cost over three heterogeneous models.

| Method | VGG-7 | | ResNet-8 | | ResNet-10 | |
|---|---|---|---|---|---|---|
| | TIME | COMM. | TIME | COMM. | TIME | COMM. |
| CRYPTFLOW2 | 48.70 | 651.51 | 56.21 | 1110.39 | 97.46 | 1395.18 |
| CHEETAH | 3.95 | 116.14 | 4.29 | 94.51 | 6.79 | 169.35 |
| CRYPTGPU | 1.61 | 144.51 | 2.02 | 131.39 | 2.79 | 221.57 |
| GUARDHFL | **0.73** | **75.52** | **0.98** | **87.60** | **1.29** | **120.26** |

Table 4: The model accuracy of three datasets in GuardHFL on different ratios of participating clients (0.6, 0.8 and 1), and querying strategies (Q-priv and Q-syn).

| Dataset | SVHN | | | CIFAR10 | | | Tiny ImageNet | | |
|---|---|---|---|---|---|---|---|---|---|
| Ratio of clients | 0.6 | 0.8 | 1 | 0.6 | 0.8 | 1 | 0.6 | 0.8 | 1 |
| Before GuardHFL | | 75.46 | | | 56.66 | | | 22.26 | |
| Q-priv | 79.43 | 79.56 | 80.29 | 60.82 | 61.01 | 61.49 | 24.89 | 25.11 | 25.23 |
| Q-syn  2.5K | 80.09 | 80.32 | 81.69 | 62.87 | 63.05 | 63.23 | 25.82 | 26.03 | 26.23 |
| 5.0K | 83.32 | 83.52 | 83.82 | 63.04 | 63.44 | 63.69 | 26.22 | 26.46 | 26.75 |
| 7.5K | 84.54 | 84.78 | 85.12 | 62.97 | 63.64 | 63.88 | 27.14 | 27.54 | 27.75 |
| 10K | 84.58 | 84.97 | 85.62 | 63.79 | 63.82 | 64.56 | 27.67 | 28.19 | 28.46 |

**Impact of Non-IID datasets.** We illustrate the impact of Non-IID data on model accuracy in Figure 7, using CIFAR10 as an example. Figures 7(a), 7(b) and 7(c) visualize the distributions of Non-IID samples among clients with different $\mathrm{Dir}(\alpha)$. When $\alpha = 100$, the distribution is close to uniform sampling. When $\alpha = 0.5$, the sample distribution of each class

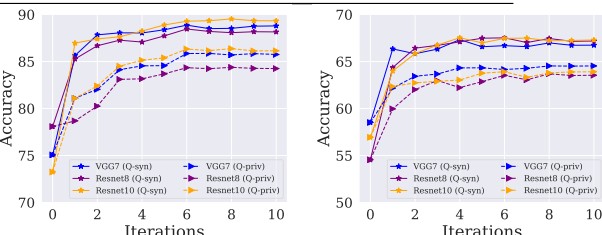

Figure 6: Accuracy curves of each heterogeneous model in GuardHFL. Left (SVHN); Right (CIFAR10)

among clients is extremely uneven. From Figure 7(d) we observe that the higher the degree of Non-IID-ness, the lower the accuracy of models. Notably, GuardHFL can still significantly improve the performance of models under the Non-IID environment.



| (a) $\alpha = 100$ | (b) $\alpha = 1$ | (c) $\alpha = 0.5$ | (d) Accuracy |
|---|---|---|---|

Figure 7: Visualization of Non-IID-ness among clients with different Dirichlet distribution $\alpha$ on CIFAR10. The size of scattered points indicates the number of training samples of that class.

**Impact of other factors.** Due to space constraints, we report other experimental results in Appendix A.1.2. In particular, Figure 9 shows the accuracy of each heterogeneous model with different numbers of query data. Figures 10(a) and 10(b) illustrate the impact of different numbers of private training data on SVHN and CIFAR10. Figure 10(c) details the impact of different query data construction methods.

## 5 CONCLUSION

We propose GuardHFL, an efficient and private HFL framework to formally provide the privacy guarantees of query samples, model parameters and prediction results. The core constructions of GuardHFL are customized multiplication and comparison protocols based on lightweight secret sharing and PRFs techniques. Extensive experiments demonstrate that GuardHFL outperforms prior art in both communication and runtime performance. We consider the following future directions. (1) The communication cost of GuardHFL, which is also the limitation of the standard HFL paradigm, will be further improved. One possible mitigation is to extend the insight of the $k$-regular graph in FL (Bell et al., 2020) to HFL, and carefully design protocols from scratch. The main idea is that in FL it is enough for each party to speak to $k < n - 1$ other parties via the server, where $n$ is the number of parties. (2) The security of GuardHFL will be improved to defeat more powerful malicious adversaries. Unfortunately, even using the best-known technique (Koti et al., 2021), the overhead will be increased by several orders of magnitude. We leave these improvements as future work.

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

# A APPENDIX

## A.1 MORE DETAILS ON EXPERIMENT EVALUATION

### A.1.1 EXPERIMENTAL SETUP

**Datasets.** We evaluate `GuardHFL` on the following standard datasets for image classification: (1) SVHN is a real-world image dataset obtained from house numbers in Google Street View images, which contains 600,000 32×32 RGB images of printed digits from 0 to 9. (2) CIFAR10 consists of 60,000 32×32 RGB images in 10 classes. There are 50,000 training images and 10,000 test images. (3) Tiny ImageNet contains 100,000 images of 200 classes downsized to 64×64 colored images. Each class has 500 training images, 50 validation images and 50 test images.

**Experimental configuration.** Each of entities, i.e., $P_Q$, $P_A$ and the server, is run on the Ubuntu 18.4 system with Intel(R) 562 Xeon(R) CPU E5-2620 v4(2.10 GHz) and 16 GB of RAM and NVIDIA 1080Ti GPU. We leverage PyTorch's torch.distributed package for communication similar as Knott et al. (2021); Tan et al. (2021). We ran our benchmarks in the LAN setting, where following Huang et al. (2022) the bandwidth is about 384 MBps and the latency is 0.3ms.

**Training procedure.** At the *local training* phase, each client trains the local model from scratch using stochastic gradient descent optimization. For SVHN, CIFAR10, and Tiny ImageNet, the loss function is cross-entropy with the learning rate of 0.5, 0.1, 0.01, respectively. Besides, the batch size is 256, 64 and 64, respectively. When the clients retrain the local model at the *local retraining* step, they use Adam optimizer for 50 epochs with learning rate of 2e-3 decayed by a factor of 0.1 on 25 epochs, where the batch size is 256 on SVHN, and 64 on both CIFAR10 and Tiny ImageNet.

### A.1.2 EXPERIMENTAL RESULTS

**Impact of GPU acceleration.** To further explore the impact of GPU acceleration, we evaluate `GuardHFL` on both CPU and GPU with different batch sizes of query data. Figure 8 reports the results of VGG-style and ResNet-style networks on CIFAR10, where the GPU-based setting is always superior to the CPU analogs. As the batch size increases, the advantage of GPU-based protocols becomes more pronounced, e.g., 14× runtime reduction on ResNet-8 over the batch size 64.

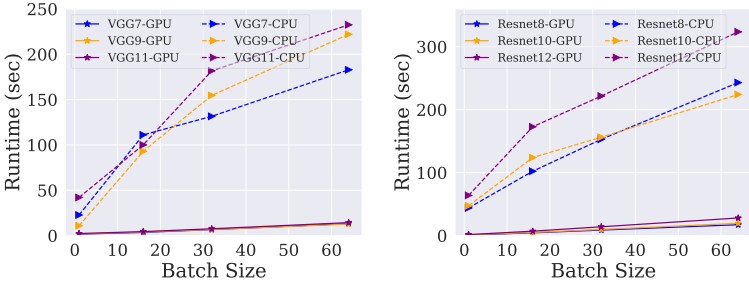

Figure 8: The runtime of `GuardHFL` under CPU/GPU with varied batch sizes of query data on CIFAR10.

**Impact of the number of query data.** Figure 9 shows the accuracy of each heterogeneous model with different numbers of query data. We observe that `GuardHFL` consistently improves the model accuracy on above datasets and heterogeneous models. Specifically, as the number of query data increases (from 2.5K to 10K), the accuracy of all three models increases by about 5%.

**Impact of the number of private training data.** Figures 10(a) and 10(b) illustrate the model accuracy of `GuardHFL` under different number of private training data on SVHN and CIFAR10. We can observe that as the number of training data increases, the model performance is on the rise. The main reason is that models can learn better on more training data and can construct more synthetic samples to query, so as to promote the transfer of model knowledge.

**Impact of query data construction strategies.** Figure 10(c) gives the model accuracy under three advanced data augmentation strategies, including cutmix (Yun et al., 2019), cutout (DeVries &

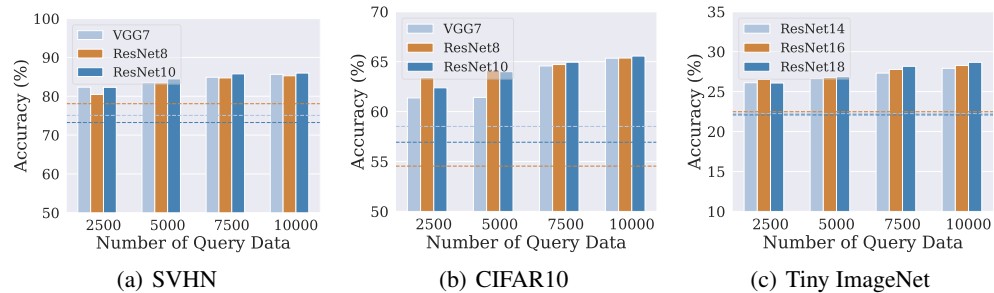

Figure 9: The accuracy of each heterogeneous model as the number of query data increases. Dashed lines represent the model accuracy before `GuardHFL`.

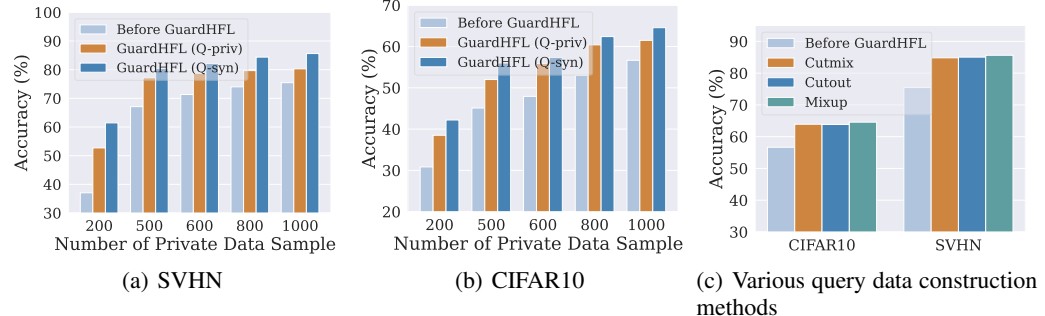

Figure 10: The model accuracy under different number of training data and query data construction methods on SVHN and CIFAR10.

Taylor, 2017), and mixup (Zhang et al., 2018). Cutmix (Yun et al., 2019) can be formulated as $\tilde{x}_{i,j} = M \cdot x_i + (1 - M) \cdot x_j$, where $M \in \{0, 1\}^{W \times H}$ is a binary mask matrix of size $W \times H$ to indicate the location of dropping out and filling from the two images $x_i$ and $x_j$. Cutout (DeVries & Taylor, 2017) augments the dataset with partially occluded versions of original samples. Mixup (Zhang et al., 2018) constructs synthetic samples by a convex combination on two images $x_i$ and $x_j$ with different coefficients $\lambda$, in which $\tilde{x}_{i,j} = \lambda \cdot x_i + (1 - \lambda) \cdot x_j$. We observe that those strategies are good choices for query data construction in `GuardHFL`.

## A.2  MORE DETAILS ON THE DESIGNED SCHEME

### A.2.1  GRAPHIC DEPICTION OF END-TO-END SECURE PREDICTION SCHEME

Figure 11 gives a graphic depiction to illustrate the end-to-end secure prediction scheme, where the input is secret-shared sample $[x]$. The whole process maintains the following *invariant*: the server and $P_A$ begin with secret shares of the input and after each layer, end with secret shares (over the same ring) of the output. Honest-but-curious security of `GuardHFL` will follow trivially from sequential composibility of individual layers. To be specific, $[x]$ first passes through a convolutional layer that can be formalized as the secure matrix multiplication operation $\omega_1[x]$ ($\omega_1$ is the parameter) using protocol $\Pi_{\text{Matmul}}$ in Figure 3. The secret-shared outputs of this layer, i.e., $[y_1]_0$ and $[y_1]_1$, are obtained by $P_A$ and the server, respectively. For the subsequent ReLU layer, protocol $\Pi_{\text{ReLU}}$ in Figure 4 is executed to return $[y_2]_0$ and $[y_2]_1$ to $P_A$ and the server respectively. Then Maxpooling on $[y_2]$ can be evaluated via protocol $\Pi_{\text{ReLU}}$ as described in Section 3.2, to outputs the secret-shared values $[y_3]_0$ and $[y_3]_1$. When the secure prediction reaches the final fully-connected layer with inputs $[y_{n-1}]_0$ and $[y_{n-1}]_1$, protocol $\Pi_{\text{Matmul}}$ is executed. In the end, $P_A$ and the server obtain the secret-shared predicted logit, i.e., $[\text{logit}]_0$ and $[\text{logit}]_1$, respectively.

### A.2.2  MORE DETAILS ON CRYPTOGRAPHIC PROTOCOLS

**Secret sharing and Beaver's multiplication protocol.** As shown in Section 2.4, `GuardHFL` utilizes the additive secret sharing primitive to protect the privacy of sensitive information. Assuming two secret-shared values are $[x]$ and $[y]$ owned by two parties $P_i$, $i \in \{0, 1\}$, addition and subtraction operations ($[z] = [x] \pm [y]$ in $\mathbb{Z}_{2^\ell}$) can be realized locally without any communication, i.e., each $P_i$

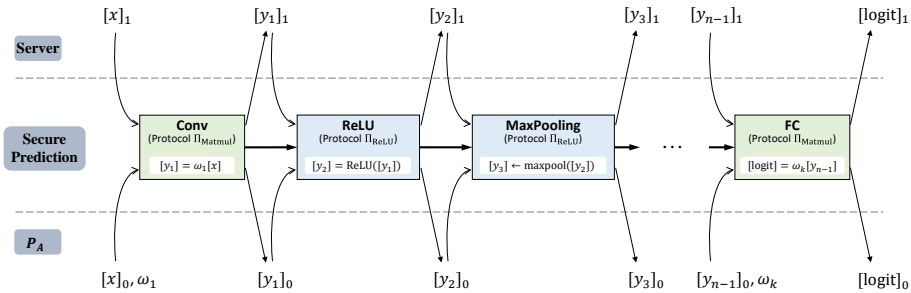

Figure 11: End-to-end secure prediction protocol. Green boxes represent linear layers (including convolutional/fully connected/Avgpooling layers), and blue boxes represent non-linear layers (including ReLU/Maxpooling layers).

computes $[z]_i = [x]_i \pm [y]_i \mod 2^\ell$. Besides, multiplication operation, i.e., $[z] = [x][y]$, is evaluated using Beaver's multiplication triples (Demmler et al., 2015), where each triple refers to $(a, b, c)$ with the constraint $c = ab$ that is generated by cryptographic techniques (Demmler et al., 2015) or a trusted dealer (Riazi et al., 2018). Specifically, the multiplication over secret-sharing values can be evaluated in the following:

$$z = xy = ([x]_0 + [x]_1)([y]_0 + [y]_1) = \overbrace{[x]_0[y]_0}^{P_0} + \overbrace{[x]_1[y]_1}^{P_1} + [x]_0[y]_1 + [x]_1[y]_0 \tag{3}$$

where for $i \in \{0, 1\}$, $[x]_i[y]_i$ can be computed locally, and $[x]_i[y]_{1-i}$ will be evaluated as follows. Taking $[x]_0[y]_1$ as an example, assuming $P_0$ and $P_1$ already hold $(a, [c]_0)$ and $(b, [c]_1)$, respectively, $P_0$ first sends $[x]_0 + a$ to $P_1$, while $P_1$ sends $[y]_1 - b$ to $P_0$. Then $P_0$ computes one share as $[x]_0([y]_0 - b) - [c]_0$, and $P_1$ computes another as $([x]_1 + a)[y]_1 - [c]_1$, locally. In this way, the outputs are still in the form of secret sharing.

**Diffie-Hellman key agreement protocol.** In `GuardHFL`, we utilize PRFs to overcome the cross-client communication limitation, where the consistent PRF seed between clients are generated using the Diffie-Hellman Key Agreement (DH) protocol (Diffie & Hellman, 1976). Note that the consistent seed between the server and the client can be directly sampled by the server and then sent to the client without the DH protocol. Figure 12 gives the secure seed generation protocol $\Pi_{\text{seed}}$. Formally, the DH protocol consists of the following three steps:

- DH.param$(k) \rightarrow (\mathbb{G}, g, q, H)$ generates a group $\mathbb{G}$ of prime order $q$, along with a generator $g$, and a hash function $H$.
- DH.gen$(\mathbb{G}, g, q, H) \rightarrow (x_i, g^{x_i})$ randomly samples $x_i \in \mathbb{Z}_q$ as the secret key and let $g^{x_i}$ as the public key.
- DH.agree$(x_i, g^{x_j}, H) \rightarrow s_{i,j}$ outputs the *seed* $s_{i,j} = H((g^{x_j})^{x_i})$.

Correctness requires that for any key pairs $(x_i, g^{x_i})$ and $(x_j, g^{x_j})$ generated by two paries $P_i$ and $P_j$ using DH.gen under the same parameters $(\mathbb{G}, g, q, H)$, DH.agree$(x_i, g^{x_j}, H) = $ DH.agree$(x_j, g^{x_i}, H)$. Besides, in `GuardHFL` security requires that for any adversary who steals $g^{x_i}$ and $g^{x_j}$ (but neither of the corresponding $x_i$ and $x_j$), the agreed secret $s_{i,j}$ derived from those keys is indistinguishable from a uniformly random value (Abdalla et al., 2001).

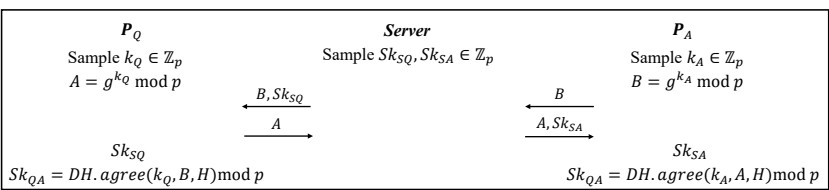

Figure 12: Secure PRF seed generation protocol $\Pi_{\text{Seed}}$

### A.2.3 DISTINGUISH GUARDHFL FROM OTHER PRIVATE SETTINGS.

GuardHFL is in line with the standard HFL paradigm (Li & Wang, 2019) with the additional benefit of privacy protection. As declared in Introduction, GuardHFL is the first-of-its-kind privacy-preserving HFL framework, which is different from existing privacy-preserving training efforts. The latter can be divided into two categories: (1) privacy-preserving federated learning (Bonawitz et al., 2017; Bell et al., 2020), and (2) secure multi-party training (Tan et al., 2021; Keller & Sun, 2022). In the following, we give a detailed analysis.

**Comparison to privacy-preserving federated learning.** In the privacy-preserving federated learning (FL), each user locally computes the gradient with his private database, and then a secure aggregation protocol is executed at the server side for aggregating the local gradients and updating the global model. However, as shown in Introduction, secure gradient aggregation cannot be realized in heterogeneous FL (HFL), due to the heterogeneity of the clients' models. Instead, GuardHFL follow a general HFL training paradigm (Li & Wang, 2019), which contains three steps: local training, querying, and local re-training. GuardHFL focuses on solving the privacy issue caused by the querying stage, and mainly proposes a query datasets generation (refer to Section 3.4) and a secure querying protocol (refer to Section 3.1 - Section 3.3).

**Comparison to secure multi-party training.** Secure multi-party training is typically an outsourced training setting, where resource-constrained clients outsource the entire training task to non-colluding multiple servers in a privacy-preserving manner. It requires a secure training protocol to finally yield a well-trained model. Different from secure multi-party training, GuardHFL enables clients to collaboratively and securely train their own customized models that may differ in size and structure. Moreover, as discussed above, the general HFL paradigm contains three steps: local training, querying and local re-training, where the local training and re-training stages are evaluated locally without revealing privacy. Therefore, the privacy-preserving HFL requires an HFL-friendly secure querying protocol (i.e., a customized inference protocol).

### A.2.4 EXTEND EXISTING 2PC PROTOCOLS TO HFL

With non-trivial adaptation, existing secure 2-party querying schemes (Mishra et al., 2020; Rathee et al., 2020; Huang et al., 2022) can be extended to the communication-limited HFL setting. However, as shown in Section 4.1, such extension introduces expensive communication and computation overheads compared with our GuardHFL. In the following we divide these schemes into three categories, i.e., pure OT-based protocols, pure HE-based protocols, and hybrid protocols, and give the corresponding extension designs.

To extend the pure OT-based secure querying protocols such as CrypTFlow2 (Rathee et al., 2020) into HFL, $P_Q$ first secret-shares query samples to the server and $P_A$ using our protocol $\Pi_{\text{Share}}$ in Section 3.1. Then the server and $P_A$ execute secure prediction based on the method proposed in Rathee et al. (2020). After that, adopting our secure aggregation protocol $\Pi_{\text{Agg}}$ in Section 3.3, the aggregated predictions will return to $P_Q$. Although the OT-based schemes can be extended to HFL by combining the designed protocols in GuardHFL, it requires too many communication rounds due to the usage of OT primitives.

To extend the pure HE-based secure querying protocols (Gilad-Bachrach et al., 2016; Lee et al., 2021) to HFL, $P_Q$ first encrypts the query samples and asks the server to pass them to $P_A$. After that, $P_A$ evaluates secure prediction non-interactively in the ciphertext environment. Then $P_A$ sends encrypted predictions to the server. The server aggregates these encrypted predictions utilizing the additive homomorphism of HE and sends the aggregated results to $P_Q$. Although it is trivial to extend the schemes equipped with the HE-based scheme to the communication-limited setting, they have two key problems: 1) activation functions need to be approximated as low-degree polynomials, which leads to serious accuracy loss; 2) due to the inherent high computation complexity, HE-based secure prediction is difficult to extend to large-scale models.

For hybrid secure querying protocols that evaluates linear layers using HE and non-linear layers using OT or GC, such as Cheetah (Huang et al., 2022), we discuss the extended algorithms for each layer separately. *For the linear layer*, 1) $P_Q$ encrypts query samples with HE and sends the

ciphertext to $P_A$ through the server[5]. 2) $P_A$ evaluates linear layers locally, and returns the encrypted masked outputs to $P_Q$ through the server. 3) $P_Q$ decrypts it to obtain the masked outputs, which are then sent to the server. As a result, the masked outputs of linear layers are secret-shared between the server and $P_A$. *For the non-linear layer*, given that the server and $P_A$ hold shares of the linear layer's outputs, the two parties invoke the OT protocols to evaluate the non-linear functions. Therefore, such extension comes at the cost of heavy computational and communication complexity.

In summary, although existing 2PC protocols can be extended to the HFL setting with the cross-communication restriction, they sacrifice the efficiency due to the lack of customized protocols and the adoption of heavy cryptographic primitives. Therefore, `GuardHFL` shows better adaptability and efficiency in the natural HFL scenarios.

## A.3 SECURITY ANALYSIS

Table 5: The ideal functionality

---

Input sharing functionality $\mathcal{F}_{\mathsf{Share}}$:

- **Input**: $P_Q$: query data $x$.
- **Output**: $P_A$: $[x]_0 \in \mathbb{Z}_{2^\ell}$. Server: $[x]_1 = x - [x]_0 \bmod 2^\ell$.

Matrix multiplication functionality $\mathcal{F}_{\mathsf{Matmul}}$:

- **Input**: Server: $[x]_1 \in \mathbb{Z}_{2^\ell}$. $P_A$: $[x]_0 \in \mathbb{Z}_{2^\ell}$, model parameter $\omega$.
- **Output**: Server: $[y]_1 \in \mathbb{Z}_{2^\ell}$. $P_A$: $[y]_0 = \omega x - [y]_1 \bmod 2^\ell$.

MSB functionality $\mathcal{F}_{\mathsf{msb}}$:

- **Input**: Server: $[x]_1 \in \mathbb{Z}_{2^\ell}$. $P_A$: $[x]_0 \in \mathbb{Z}_{2^\ell}$.
- **Output**: Server: $[\mathsf{msb}(x)]_1^B \in \mathbb{Z}_2$. $P_A$: $[\mathsf{msb}(x)]_0^B = \mathsf{msb}(x) \oplus [\mathsf{msb}(x)]_1^B \bmod 2$.

ReLU functionality $\mathcal{F}_{\mathsf{ReLU}}$:

- **Input**: Server: $[x]_1 \in \mathbb{Z}_{2^\ell}$. $P_A$: $[x]_0 \in \mathbb{Z}_{2^\ell}$.
- **Output**: Server: $[y]_1 \in \mathbb{Z}_{2^\ell}$. $P_A$: $[y]_0 = \mathsf{ReLU}(x) - [y]_1 \bmod 2^\ell$.

Maxpooling functionality $\mathcal{F}_{\mathsf{Maxpool}}$:

- **Input**: Server: $[x]_1 \in \mathbb{Z}_{2^\ell}$. $P_A$: $[x]_0 \in \mathbb{Z}_{2^\ell}$.
- **Output**: Server: $[y]_1 \in \mathbb{Z}_{2^\ell}$. $P_A$: $[y]_0 = \mathsf{Maxpool}(x) - [y]_1 \bmod 2^\ell$.

Result aggregation functionality $\mathcal{F}_{\mathsf{Agg}}$:

- **Input**: Server: $[x_i]_1 \in \mathbb{Z}_{2^\ell}$, $i \in [C]$. $P_A^i$: $[x_i]_0 \in \mathbb{Z}_{2^\ell}$.
- **Output**: $P_Q$: $y = \mathsf{softmax}(\sum_{i=1}^{|C|} x_i)$.

---

Intuitively, `GuardHFL` only reveals the aggregated prediction to $P_Q$ without the responding parties' model parameters, and the server and $P_A$ learn zero information about the querying parties' data. This is because all intermediate sensitive values are secret-shared. Next, we give a formal analysis.

Our security proof follows the standard ideal-world/real-world paradigm (Canetti, 2001): in the real world, three parties (i.e., the server, $P_Q$, and $P_A$) interact according to the protocol specification, and in the ideal world, they have access to an ideal functionality shown in Table 5. When a protocol invokes another sub-protocol, we use the $\mathcal{F}$-hybrid model for the security proof by replacing the sub-protocol with the corresponding functionality. Note that our proof works in the $\mathcal{F}_{\mathsf{PRF}}$-hybrid model where $\mathcal{F}_{\mathsf{PRF}}$ represents the ideal functionality corresponding to the PRF protocol. The executions in both worlds are coordinated by the environment Env, who chooses the inputs to parties and plays

---

[5]To be more precise, this step is for the input layer. In the hidden layer, one of the input shares of the linear layer should be encrypted by the server and sent to $P_A$.

the role of a distinguisher between the real and ideal executions. We will show that the real-world distribution is computationally indistinguishable to the ideal-world distribution.

**Theorem A.1.** $\Pi_{\mathsf{Share}}$ *securely realizes the functionality* $\mathcal{F}_{\mathsf{Share}}$ *in the* $\mathcal{F}_{\mathsf{PRF}}$-*hybrid model.*

*Proof.* Note that $P_Q$ and $P_A$ receive no messages in $\Pi_{\mathsf{Share}}$, and hence the protocol is trivially secure against corruption of $P_Q$ and $P_A$. Next, the only message that the server receives is the value $[x]_1$. However, $[x]_1 = x - r$, where given the security of PRF, $r$ is a random value unknown to the server. Thus, the distribution of $[x]_1$ is uniformly random from the server's view and the information learned by the server can be simulated. $\square$

**Theorem A.2.** $\Pi_{\mathsf{Matmul}}$ *securely realizes the functionality* $\mathcal{F}_{\mathsf{Matmul}}$ *in the* $\mathcal{F}_{\mathsf{PRF}}$-*hybrid model.*

*Proof.* Note that $P_Q$ receives no messages in $\Pi_{\mathsf{Matmul}}$, and hence the protocol is trivially secure against corruption of $P_Q$. The only message that $P_A$ receives is the value $[x]_1 - b$. However, given the security of PRF, $b$ is a random value unknown to $P_A$. Thus, the distribution of $[x]_1 - b$ is computationally indistinguishable from a uniformly random distribution in $P_A$'s view, and the information learned by $P_A$ can be simulated. Next, during the protocol, the server learns $[c]_1$ and $w + a$. However, the distribution of $[c]_1$ and $w + a$ is computationally indistinguishable from a uniformly random distribution in the server's view, since given the security of PRF, $a$ and $[c]_1$ are random values unknown to the server. Thus, the information learned by the server can be simulated. $\square$

**Theorem A.3.** $\Pi_{\mathsf{ReLU}}$ *securely realizes the functionality* $\mathcal{F}_{\mathsf{ReLU}}$ *in the* $(\mathcal{F}_{\mathsf{Matmul}}, \mathcal{F}_{\mathsf{msb}})$-*hybrid model.*

*Proof.* Note that as shown in Section 3.2, $\Pi_{\mathsf{ReLU}}$ consists of $\Pi_{\mathsf{msb}}$ and $\Pi_{\mathsf{Matmul}}$. Therefore, the ReLU protocol is trivially secure in the $(\mathcal{F}_{\mathsf{Matmul}}, \mathcal{F}_{\mathsf{msb}})$-hybrid model. $\square$

**Theorem A.4.** $\Pi_{\mathsf{Maxpool}}$ *securely realizes the functionality* $\mathcal{F}_{\mathsf{Maxpool}}$ *in the* $\mathcal{F}_{\mathsf{ReLU}}$-*hybrid model.*

*Proof.* As shown in Section 3.2, $\Pi_{\mathsf{Maxpool}}$ consists of several invocations of $\Pi_{\mathsf{ReLU}}$. Therefore, the protocol $\Pi_{\mathsf{Maxpool}}$ is trivially secure in the $\mathcal{F}_{\mathsf{ReLU}}$-hybrid model. $\square$

**Theorem A.5.** $\Pi_{\mathsf{Agg}}$ *securely realizes the functionality* $\mathcal{F}_{\mathsf{Agg}}$ *in the* $\mathcal{F}_{\mathsf{PRF}}$-*hybrid model.*

*Proof.* Note that $P_A$ receives no messages in $\Pi_{\mathsf{Agg}}$, and hence the aggregation protocol is trivially secure against the corruption of $P_A$. Next, the only message that the server receives is the value $[x_i]_0 - r_i$. However, given the security of PRF, $r_i$ is a random value unknown to the server. Thus, the distribution of $[x_i]_0 - r_i$ is computationally indistinguishable from a uniformly random distribution in the server's view and the information learned by the server can be simulated. After the aggregation, $P_Q$ only learns the aggregated result $\sum_{i \in [C]} x_i$, but is unknown to each $x_i$. Therefore, the aggregation protocol is secure assuming the aggregation result will not reveal privacy. $\square$

## A.4 RELATED WORK

### A.4.1 HETEROGENEOUS FEDERATED LEARNING

Federated learning (FL) achieves collaboration among clients via sharing model gradients. While successful, it still faces many challenges, among which, of particular importance is the heterogeneity that appear in all aspects of the learning process. This consists of *model heterogeneity* (Li & Wang, 2019) and *statistical heterogeneity* (Zhu et al., 2021). Statistical heterogeneity means that parties' data comes from distinct distributions (i.e., Non-IID data), which may induce deflected local optimum. Solving the statistical heterogeneity has been extensively studied, such as Dinh et al. (2020); Zhu et al. (2021); Yurochkin et al. (2019); Fallah et al. (2020); Yoon et al. (2021), and is out of the scope of this work. However, GuardHFL may help alleviate the statistical heterogeneity due to the customized model design and the knowledge distillation-based aggregation rule.

Our work mainly focuses on the model heterogeneity that has been explored in recent works (Li & Wang, 2019; Lin et al., 2020; Choquette-Choo et al., 2021). In particular, Li & Wang (2019) proposed the first FL framework FedMD supporting heterogeneous models by combining transfer learning and knowledge distillation techniques. They first used a public dataset to pre-train the model

Table 6: Comparison with prior works on properties necessary for federated learning

| Framework | Privacy | | Usability | | Efficiency | |
|---|---|---|---|---|---|---|
| | Data Privacy | Model Privacy | Model Heterogeneity | w/o Dataset Dependency | GPU Compatibility | Protocol Efficiency |
| Bonawitz et al. (2017) | ✓ | ✗ | ✗ | ✓ | ✗ | ✓ |
| Bell et al. (2020) | ✓ | ✗ | ✗ | ✓ | ✗ | ✓ |
| Sav et al. (2021) | ✓ | ✓ | ✗ | ✓ | ✗ | ✗ |
| Jayaraman & Wang (2018) | ✓ | ✗ | ✗ | ✓ | ✗ | ✓ |
| Li & Wang (2019) | ✗ | ✓ | ✓ | ✗ | ✓ | - |
| Choquette-Choo et al. (2021) | ✓ | ✓ | ✓ | ✗ | ✗ | ✗ |
| Lin et al. (2020) | ✗ | ✗ | ✓ | ✗ | ✓ | - |
| Sun & Lyu (2021) | ✗ | ✓ | ✓ | ✗ | ✓ | ✓ |
| Diao et al. (2021) | ✗ | ✗ | ✓ | ✓ | ✓ | - |
| **GuardHFL** | ✓ | ✓ | ✓ | ✓ | ✓ | ✓ |

and transferred to the task of private datasets. After that, to exchange the knowledge, each party used the public data and the aggregated predictions from others as carrier for knowledge distillation. To further improve model accuracy, Lin et al. (2020) proposed FedDF, similar to FedMD, which also used the model distillation technique for knowledge sharing. The difference is that they first performed FedAvg on parties' local models and integrated knowledge distillation on the aggregated model. The dependence on model averaging leads to limited model heterogeneity. Besides, Diao et al. (2021) focused on heterogeneous parties equipped with different computation and communication capabilities. In their framework, each party only updated a subset of global model parameters through varying the width of hidden channels, which reduces the computation and communication complexity of local models. However, this approach only learns a single global model, rather than unique models designed by parties. Moreover, as described in Introduction, HFL suffers from several privacy issues, which are not considered in the above works. The first is the direct leakage of querying samples. The task-related querying samples may contain private information such as disease diagnosis in healthcare. If the querying party directly delivers query samples to responding parties for prediction via the server, such sensitive information will be leaked. The second is the implicit disclosure based on inference attacks. Given black-box access to a model, adversaries can infer the membership (Salem et al., 2019) and attribute information (Ganju et al., 2018) of the target sample or even reconstruct the original training data (Yang et al., 2019). For example, based on the prediction $y$ of the querying sample $x$ from the responding party $P_A$, the querying party can launch the membership inference attack (Salem et al., 2019) to infer whether $x$ is a training sample of $P_A$. To address the privacy concern, `GuardHFL` provide end-to-end privacy-preserving HFL services.

The privacy-preserving techniques (i.e., secure aggregation) have been studied in federated learning (Bonawitz et al., 2017; Bell et al., 2020; Sav et al., 2021; Jayaraman & Wang, 2018). However, these techniques can not be directly extended to privacy-preserving HFL. More recently, Sun & Lyu (2021) proposed a noise-free differential privacy solution for HFL to guarantee each party's privacy. However, as shown in Jayaraman & Evans (2019), there is a huge gap between the upper bounds on privacy loss analyzed by advanced mechanisms and the effective privacy loss. Thus, differentially private mechanisms offer undesirable utility-privacy trade-offs. To further formally guarantee the privacy, Choquette-Choo et al. (2021) leveraged hybrid cryptographic primitives to realize confidential and private collaborative learning. Specifically, parties learn from each other collaboratively utilizing a secure inference strategy based on 2PC and HE protocols and a private aggregation method. As noted in the Introduction, CaPC's use of heavy cryptography leads to significant efficiency and communication overheads.

In summary, we give a comparison between prior FL works and `GuardHFL` in Table 6.

### A.4.2 SECURE NEURAL NETWORK PREDICTION

Since secure prediction is a critical component of `GuardHFL`, we briefly introduce its recent progress. Neural networks present a challenge to cryptographic protocols due to their unique struc-

ture and exploitative combination of linear computations and non-linear activation functions. In real scenarios, model prediction can be viewed as a two-party computation case, where one party with private query data wants to obtain prediction results from the other party who owns the model. During the whole process, the cryptographic protocols, typically HE and secure multi-party computation (MPC), are applied to ensure the confidentiality of the private data and model parameters.

Many existing works (Boemer et al., 2019b; Gilad-Bachrach et al., 2016; Brutzkus et al., 2019) support pure HE protocols for secure predictions. Typically, nGraph-HE (Boemer et al., 2019b;a) allows linear computations using the CKKS HE scheme. However, since a solution that builds upon HE protocols should be restricted to compute low degree polynomials, the non-polynomial activation functions, such as Maxpooling and ReLU, are forced to be evaluated in the clear by the party who owns private query data. This leaks the feature maps, from which adversaries may deduce the model weights. To solve this problem, Gilad-Bachrach et al. (2016) and Chen et al. (2019) use low-degree polynomial approximation to estimate non-linear functions. Unfortunately, it will affect the inference accuracy, while leading to huge computation overhead.

On the other hand, several libraries (Mohassel & Zhang, 2017; Knott et al., 2021; Wagh et al., 2019) employ primarily MPC techniques in secure predictions, which provide linear and non-linear protocols through the usage of oblivious transfer (OT), garbled circuit (GC) and secret sharing. In particular, CryptTen (Knott et al., 2021) performs linear operations based on $n$-out-of-$n$ additive secret sharing over the ring $\mathbb{Z}_{2^l}$. However, it requires a trusted third party to assist the secure prediction process, which is unrealistic in the real-world setting. CrpytGPU (Tan et al., 2021) builds on CrypTen, working in the 3-party setting using *replicated secret shares*. Although the scalability is poor, it introduces an interface to losslessly embed cryptographic operations over secret-shared values in a discrete somain into floating-point operations, which can implement the whole inference process on the GPU. Recently, Keller & Sun (2022) proposed a secure quantized training protocol that outperforms CryptGPU in the cryptographic performance. Unfortunately, this work cannot be applied in HFL and is not comparable to `GuardHFL`. The main reasons are: (1) `GuardHFL` and Keller & Sun (2022) are concerned with completely different tasks. `GuardHFL` builds on the standard HFL setting, where multiple parties collaboratively train individual models with the assistance of a server. Keller & Sun (2022) focuses on the outsourced training scenario, i.e., multiple servers jointly execute standard model training algorithm to obtain a well-trained model. (2) Moreover, the protocols in Keller & Sun (2022) are designed for a specific network architecture, i.e., quantized neural networks, which cannot be applied to the general models in `GuardHFL`. Therefore, Keller & Sun (2022) and `GuardHFL` are two fully orthogonal works.

Many other works focus on hybrid protocols, in which they combines the advantages of HE and MPC to improve prediction efficiency (Juvekar et al., 2018; Mishra et al., 2020; Rathee et al., 2020; Huang et al., 2022). CrypTFlow2 (Rathee et al., 2020) implements two class of protocols, HE-based and OT-based, for linear operations. For non-linear layers, they also design efficient protocols based on OT. Cheetah (Huang et al., 2022) improves CrypTFlow2 with customized HE-based linear protocols and improved OT-based non-linear protocols. HE-transformer (Boemer et al., 2019a) employs nGraph-HE for evaluation of linear operations, and ABY framework for GC to evaluate non-linear functions (Demmler et al., 2015). Since non-linear operations cannot be parallelized between query data, GC is inefficient, especially for large networks with thousands of parameters. In contrast, our `GuardHFL` avoids the use of heavy cryptographic tools like HE and OT, and only employs secret sharing and PRFs to achieve high efficiency, confidentiality and practicability.

