# OpenReview forum: "GuardHFL: Privacy Guardian for Heterogeneous Federated Learning"
_ICLR.cc/2023/Conference — Submitted to ICLR 2023_

### Official Review · Reviewer_q3RN · 2022-10-20

**Confidence:** 4
**Correctness:** 2
**Technical Novelty And Significance:** 3
**Empirical Novelty And Significance:** 2
**Recommendation:** 3

**Clarity, Quality, Novelty And Reproducibility:**

I don't understand the notion of the auxiliary querying dataset. Where does it come from? I can't be entirely randomly generated, as that wouldn't make any sense in many of the considered applications (say MNIST, ImageNet). So it must be somewhat real data, which comes with privacy implications.

The caption of Table 3 should make it clear that the times are for inference (I assume).

I think it's not fair to highlight that this work outperforms the state of the art by two orders of magnitude. Cheetah improves more than ten-fold over CryptFlow2 in the same security setting, so the latter is clearly not the state of the art. Furthermore, the setting of two parties plus an auxiliary party is more comparable to the three-party setting of CryptGPU than the pure two-party setting of CryptFlow2 and Cheetah, so only CryptGPU can be considered an appropriate baseline. Furthermore, CryptGPU itself has been outperformed by the work of Keller and Sun (ICML'22) at least when it comes to training.


**Strength And Weaknesses:**

This is an interesting idea, but not enough attention is given to privacy issues regarding the dataset used for local training (line 13 in Algorithm 1). If this dataset is useful to the model it intuitively must leak something about the other parties' datasets, and if it doesn't, how can it improve the model? In light of that, I don't think it's entirely fair to compare to prior work where the training happens completely in private.


**Summary Of The Paper:**

The paper presents a privacy-preserving federated learning protocol. Unlike prior works, it avoids running the training in a privacy-preserving manner but instead uses local training on sample-answer pairs that have been obtained using private inference.


**Summary Of The Review:**

Interesting idea with too many open questions in the execution

---

> ### Author Response · Authors · 2022-11-15
> **Response to Reviewer q3RN (1/2)**
>
> Dear Reviewer,
>
> Thanks for your valuable comments. We address each of your points below.
>
> ***Q1.** The paper presents a privacy-preserving federated learning protocol. Unlike prior works, it avoids running the training in a privacy-preserving manner but instead uses local training on sample-answer pairs that have been obtained using private inference.*
>
> **A1.** Thanks for your comment. We would like to clarify the main differences between our GuardHFL and existing privacy-preserving training works.
> The latter can be divided into two categories: 1) privacy-preserving federated learning [1] [2], and 2) secure multi-party training such as CryptGPU and ICML'22 [3].
> We give a detailed analysis below, and have added a discussion in Introduction and Appendix A.2.3 of the revision to make our contribution clear.
>
> **Comparison to privacy-preserving federated learning.**
> In the privacy-preserving federated learning (FL), each user locally computes the gradient with his private database, and then a secure aggregation protocol is executed at the server side for aggregating the local gradients and updating the global model.
> However, as shown in Introduction of the manuscript, secure gradient aggregation can not be realized in heterogeneous FL (HFL), due to the heterogeneity of the clients’ models.
> Instead, we follow a general HFL training paradigm [4], which contains three steps: local training, querying, and local re-training.
> This work focuses on solving the privacy issue caused by the querying stage, and mainly proposes a querying data generation method (Section 3.4 and Answer 2 below) and a secure querying protocol (Section 3.1-3.3).
>
> **Comparison to secure multi-party training.**
> Secure multi-party training is typically an outsourced training setting, where  resource-constrained clients outsource the entire training task to multiple non-colluding servers in a privacy-preserving manner.
> It requires a secure training protocol to finally yield a well-trained model.
> Different from secure multi-party training, GuardHFL enables clients to collaboratively and securely train their own customized models that may differ in size and structure.
> Moreover, as discussed above, the general HFL paradigm contains three steps: local training, querying and local re-training.
> The training and re-training processes inherently are local operations without any privacy concerns.
> Therefore, this work aims to design a customized secure querying protocol for privacy-preserving HFL.
>
> ***Q2.** Where does the auxiliary querying dataset come from and are there any privacy issues?*
>
> **A2.** We apologize for possible misunderstandings.
> We would like to clarify that **no privacy breaches occur in the generation of auxiliary querying samples.**
> The querying samples are constructed locally by the querying party based on his training data using advanced data augmentation strategies (please refer to Section 3.4 of the manuscript), and no other parties and their private datasets are involved in this process.
> Therefore, the querying data construction does not reveal any private information.
> Below we introduce the detailed method for querying data consturction, and have revised Section 3.4 and Algorithm 1 to make the description clear.
>
>
> For example, given two individual training samples $x_i$ and $x_j$, the querying sample $x_{i,j}$ can be generated as $x_{i,j} = \lambda \cdot x_{i}+(1-\lambda) \cdot x_{j}$, i.e., a convex combination on $x_i$ and $x_j$ with different coefficients $\lambda$ [5].
> Please refer to Section 3.4 and Appendix A.1.2 in the manuscript for more details.
>
> ***Q3.** The caption of Table 3 should make it clear that the times are for inference (I assume).*
>
> **A3.** Thanks for your comment. We have modified the caption of Table 3 in the revision, i.e., comparison with advanced secure prediction protocols on runtime (sec) and communication (MB) cost over three heterogeneous models.
>
> ***Q4.**  I think it's not fair to highlight that this work outperforms the state of the art by two orders of magnitude. Cheetah improves more than ten-fold over CryptFlow2 in the same security setting, so the latter is clearly not the state of the art.*
>
> **A4.** Thanks for your comment. We have revised the description of the performance improvement in the revision.
>
> [1] Practical Secure Aggregation for Privacy-Preserving Machine Learning. [CCS, 2017].
>
> [2] Secure Single-Server Aggregation with (Poly)Logarithmic Overhead. [CCS, 2020].
>
> [3] Secure Quantized Training for Deep Learning. [ICML, 2022].
>
> [4] Fedmd: Heterogenous federated learning via model distillation. [NeurIPS Workshop, 2019].
>
> [5] mixup: Beyond empirical risk minimization. [ICLR, 2018].

---

> > ### Comment · Reviewer_q3RN · 2022-12-01
> > **A1/A3**
> >
> > The authors differentiate their work from training fully done in SMPC. However, they use the secure inference as a baseline in Table 3. There are many more works on secure inference, see https://dl.acm.org/doi/abs/10.1145/3407023.3407045 for an overview. Among these, https://eprint.iacr.org/2019/131 provides a figure of 4.7 seconds for ResNet-50 with three parties. I think this would be competitive with the figure of 1 second for ResNet-10 in this work.

---

> > > ### Author Response · Authors · 2022-12-02
> > > **Response to A1/A3**
> > >
> > > Thanks to the reviewer for bringing these efforts [6, 7] into our attention.
> > > Before conducting this work, we investigated the existing secure inference schemes and also found [6, 7]. We choose as baselines the most efficient two-party (Cheetah) and three-party (CryptGPU) solutions, which outperform prior works including [7]. Below, we give detailed reasons for not comparing with [7] from the perspective of model architectures and efficiency advantages:
> > >
> > >
> > > (1) GuardHFL and our baselines (Cheetah and CryptGPU) are designed for general neural network models, however [7] focuses on a specific model architecture, i.e., quantized neural networks.
> > > Thus, [7] cannot be extended to the standard HFL setting.
> > >
> > >
> > >
> > > (2) Under the same communication setting, GuardHFL outperforms [7] by a large margin in terms of the runtime and communication costs.
> > > In particular, *for the runtime cost*, given the same communication setting as ours, the runtime of [7] is 32.6 seconds for ResNet-50, instead of 4.7 seconds, from Table 8 of Cheetah (Note that GuardHFL shares the same communication setting as Cheetah, i.e., the bandwidth and latency of 384 MBps and 0.3ms, respectively.).
> > > *For the communication cost*, as shown in Table 8 of Cheetah, the communication overhead of Cheetah is much lower than that of [7].
> > > Meanwhile, as reported in Table 3 of GuardHFL, the communication performance of GuardHFL is significantly better than that of Cheetah.
> > > Therefore, GuardHFL can achieve significant performance improvements in terms of runtime and communication over [7].
> > >
> > > [6] MP2ML: a mixed-protocol machine learning framework for private inference. https://dl.acm.org/doi/abs/10.1145/3407023.3407045. [ARES, 2020].
> > >
> > > [7] Secure evaluation of quantized neural networks. https://eprint.iacr.org/2019/131. [PETs, 2020].

---

> > ### Comment · Reviewer_q3RN · 2022-12-01
> > **A2**
> >
> > Thank you for clarifying this.

---

> ### Author Response · Authors · 2022-11-15
> **Response to Reviewer q3RN (2/2)**
>
> ***Q5.** The setting of two parties plus an auxiliary party is more comparable to the three-party setting of CryptGPU than the pure two-party setting of CryptFlow2 and Cheetah, so only CryptGPU can be considered an appropriate baseline.*
>
> **A5.** Thanks for your comment.
> We would like to clarify that although the system model of GuardHFL can be formalized as three entities (i.e., $P_Q$, the server and $P_A$), it cannot be regarded as a regular three-party protocol due to the communication limitation between $P_Q$ and $P_A$.
> The reason is that as shown in Section 2.3 of our manuscript, the clients in real-world applications (i.e., $P_Q$ and $P_A$ in our work) are generally widely distributed and cannot establish direct
> communication channels with others.
> Therefore, GuardHFL is more of a two-party protocol since the communication occurs either between $P_Q$ and the server or between $P_A$ and the server.
> Actually, existing 2PC schemes such as Cheetah and CrypTFlow2 can be extended to instantiate HFL with adaptive protocol modifications (refer to Appendix A.2.4).
> Thus we give the comparison with two-party secure inference schemes, which is reasonable and fair.
>
> Moreover, as shown in Section 2.3 of our manuscript, extending 3PC protocols (e.g. CryptGPU) to HFL is non-trivial.
> However, since GryptGPU is one of the most advanced protocols, we also compare with it assuming no communication limitation.
> We would like to mention that despite such an unfair comparison, our GuardHFL still has performance advantages.
>
> ***Q6.** CryptGPU itself has been outperformed by the work of Keller and Sun (ICML'22) [3] at least when it comes to training.*
>
> **A6.** We would like to give two reasons below to illustrate that this work is not comparable to our GuardHFL in perspective of focused tasks and proposed techniques. For completeness, we have discussed [3] in Appendix A.4.2 in the revision.
>
>
> (1) Focused tasks. GuardHFL and [3] are concerned with completely different tasks.
> GuardHFL builds on the standard HFL setting, where multiple parties collaboratively train individual models with the assistance of a server.
> As shown in Answer 1, [3] focuses on the outsourced training scenario, i.e., multiple servers jointly execute standard model training algorithm to obtain a well-trained model.
> (2) Proposed techniques. The protocols in [3] are designed for a specific network architecture, i.e., quantized neural networks, which cannot be applied to the general models in GuardHFL.
> Therefore, [3] and GuardHFL are two fully orthogonal works and are not comparable.

---

> > ### Comment · Reviewer_q3RN · 2022-12-01
> > **A5**
> >
> > I remain of the opinion that three parties make a three-party protocol even if two of them don't communicate with each other. Real two-party computation is known to require primitives like oblivious transfer and homomorphic encryption, which are used by CaPC. I thus consider the comparison to said work as unfair.

---

> > > ### Author Response · Authors · 2022-12-02
> > > **Response to A5**
> > >
> > > Thanks for your reply.
> > > As the reviewer noted, GuardHFL can indeed be regarded as a communication-limited three-party protocol.
> > > However, there is currently no general three-party cryptographic protocol for this communication-limited setting.
> > > Even worse, extending existing three-party protocols to this setting requires non-trivial efforts.
> > > We achieve the privacy-preserving HFL by customizing efficient protocols for our GuardHFL framework.
> > > The main reason for comparing with existing two-party protocols is that they can be directly extended to the HFL scenario.
> > > Moreover, we also compare GuardHFL with the state-of-the-art three-party scheme (CryptGPU) in Table 3, assuming there is no the above communication limitation in CryptGPU.
> > > These experimental evaluations show that our GuardHFL significantly outperforms both the most efficient two-party and three-party protocols.

---

> ### Author Response · Authors · 2022-12-01
> **Response to Reviewer q3RN**
>
> Dear Reviewer,
>
> Thanks again for your comments! As the discussion period will end soon, could you please kindly check our responses and revisions? We believe that our responses and revisions have addressed your misunderstandings and concerns.
>
> Best regards,
>
> Authors

---

> ### Author Response · Authors · 2022-12-06
> **Response to Reviewer q3RN**
>
> Dear Reviewer,
>
> Thank you again for your valuable comments! Since the discussion period is approaching its end, we wonder if you have any further question. If so, please raise them and we're ready to clarify further any time.
>
> With respects,
>
> Authors

---

> ### Author Response · Authors · 2022-12-10
> **Response to Reviewer q3RN**
>
> Dear Reviewer,
>
> Thanks again for your comments and interactive discussion. Resolving your concerns also makes our work clearer. We believe that during this process our responses and revisions have well addressed your concerns.
>
> (1) Your main concern is the privacy implication about query samples. As you replied, we have clarified the privacy issue in our response and have addressed your concern.
>
> (2) You suggest that we should compare with the protocols under the three-party setting. In our manuscript, we have compared our GuardHFL with the state-of-the-art three-party scheme in Table 3. These experimental evaluations show that GuardHFL significantly outperforms the most efficient three-party protocol.
>
> (3) You bring some works into our attention. Before conducting this work, we investigated the existing secure inference schemes and also found them. We have given sufficient evidence in the response and revision to demonstrate the superiority of our GuardHFL compared to these efforts.
>
> Therefore, we believe that all doubts have been well resolved. Since the discussion period is approaching its end, we wonder if you have any further question. If so, please raise them and we're ready to clarify further any time. If not, would you consider raising the score of our paper appropriately?
>
> Best regards,
>
> Authors

---

### Official Review · Reviewer_Gysq · 2022-10-25

**Confidence:** 2
**Correctness:** 4
**Technical Novelty And Significance:** 3
**Empirical Novelty And Significance:** 3
**Recommendation:** 6

**Clarity, Quality, Novelty And Reproducibility:**

I think this paper is well-written. I am not an expert on this topic so I could not provide a fair judgment on the novelty.

**Strength And Weaknesses:**

Strengths: 1. Compared to prior works with no privacy protection or adaptation of existing frameworks, this proposed framework is efficient and customized.
Weaknesses: 1. Secure model predictions requires customized design for particular model architectures. I am wondering how to deal with some privacy-sensitive layers, like for example, batch normalization.
2. It would be better if this paper could provide some attacks from an adversary perspective to demonstrate privacy.

**Summary Of The Paper:**

This paper provides a practical framework for heterogeneous federated learning (HFL). Existing HFL frameworks suffer from privacy leakage through query data communication. Compared to prior work in HFL or adaptation with privacy-preserving feature, the proposed framework 1) is lightweight and efficient, 2) provides formal privacy guarantees. The key idea is to use several cryptographic primitives to perform secure query-data sharing, secure model prediction and secure result aggregation. This paper demonstrates the efficiency through extensive experiments.

**Summary Of The Review:**

I think this paper provides an efficient and privacy-preserving framework. The effectiveness/ efficiency is demonstrated through extensive experiments

---

> ### Author Response · Authors · 2022-11-15
> **Response to Reviewer Gysq**
>
> Dear Reviewer,
>
> Thanks for your valuable comments. We address your questions below.
>
> ***Q1.** I am wondering how to deal with some privacy-sensitive layers, like for example, batch normalization.*
>
> **A1.** We apologize for the lack of description of batch normalization and the possible confusion. In fact, our framework fully supports the batch normalization layer, which is also used in our evaluation such as Resnet.
> Generally, batch normalization can be formalized as a linear operation, and hence be directly implemented using our secure matrix multiplication protocol (Figure 3 of the manuscript). We have added it in Section 3.2 of the revision, and show the protocol detail as follows.
>
> The BN layer takes as input a 3-dimension tensor $\mathbf{T} \in$ $\mathbb{R}^{C \times H \times W}$ and outputs a 3-dimension tensor $\mathbf{T}^{\prime}$ of the same shape.
> This layer is specified by the tuple $(\mu, \theta)$ where $\mu \in \mathbb{R}^C$ is the scaling vector and $\theta \in \mathbb{R}^C$ is the shift vector.
> For all $c \in [C], h \in[H]$ and $w \in[W], \mathbf{T}^{\prime}$ is computed via $ \mathbf{T}^{\prime}[c, h, w]=\mu[c] \mathbf{T}[c, h, w]+\theta[c]$.
> In the secure querying, $(\mu, \theta)$ is held by the server and $\mathbf{T}$ is secret-shared between the server and the  responding party, where $\mu \mathbf{T} + \theta$ can be naturally evaluated by the secure multiplication protocol. After that, the output of the BN layer is secret-shared between the server and the responding party.
>
> ***Q2.** It would be better if this paper could provide some attacks from an adversary perspective to demonstrate privacy.*
>
> **A2.** Thanks for your comment. In the revision, we have added the privacy attacks for non-private HFL from an adversary perspective in Appendix A.4.1, and have further enhanced the motivation for privacy protection in Introduction.
> Below, we detail the two main privacy concerns.
>
> Specifically, (1) the first is the direct leakage of querying samples.
> The task-related querying samples may contain private information such as disease diagnosis in healthcare.
> If the querying party directly delivers query samples to responding parties for prediction via the server, such sensitive information will be leaked.
> (2) The second is an implicit disclosure based on inference attacks.
> Given the prediction $y$ of the querying sample $x$ from the responding party $P_A$, the querying party can launch privacy inference attacks, for instance, membership attacks [1] to infer whether $x$ is a training sample of $P_A$.
> To address the above concerns, GuardHFL provide an end-to-end privacy-preserving HFL framework.
>
> [1] Ml-leaks: Model and data independent membership inference attacks and defenses on machine learning models. [NDSS, 2019].

---

### Official Review · Reviewer_VwDD · 2022-10-28

**Confidence:** 2
**Correctness:** 3
**Technical Novelty And Significance:** 3
**Empirical Novelty And Significance:** 3
**Recommendation:** 6

**Clarity, Quality, Novelty And Reproducibility:**

The paper is well written. I would encourage the authors to release their code.


**Strength And Weaknesses:**


Improving the privacy protection techniques in FL when clients have different model architectures is an important and timely topic.

That being said, the motivation of the proposed method for federated learning does not seem to be super strong. The local training, querying and retraining approach is taken as granted. However, it is not clear to me that this should be the default, or even a popular algorithm for so-called heterogeneous FL. The authors cited (Li & Wang, 2019; Zhu et al., 2021), but the approaches in the two papers seem to be different from Figure 1 (without considering the encryption). The experiments compared with other encryption methods in collaborative learning. I would encourage the authors to think about ways to justify the training paradigm, and make stronger connections between the training paradigm and the proposed method.

I would also encourage the authors to discuss more about the application setting, and limitation of the proposed method.GuardHFL assumes the communication between clients and server are stable with possibly multiple communication targeted specific clients (query and answer), which seems to be only applicable to the  cross-silo FL, not cross-device FL. Algorithm 1 line 4-14 also seems to suggest a large communication cost.



**Summary Of The Paper:**

This paper proposed a secure querying scheme for heterogeneous federated learning (HFL). HFL is a setting where clients in collaborative learning will train a model with different model architectures. Hence the global model cannot be directly aggregated/averaged from local client models. The proposed GuardHFL can query models on other clients to get predictions of their private data without publicly sharing the query data by multi-party encryption between Query Client, Answer Client and the Server. Experiments on r SVHN, CIFAR10, and Tiny ImageNet show the efficiency of GuardHFL.


**Summary Of The Review:**

I unfortunately lack the expertise to evaluate the encryption method itself. The HFL motivation is interesting, but can benefit from more justification.

---

> ### Author Response · Authors · 2022-11-15
> **Response to Reviewer VwDD (1/2)**
>
> Dear Reviewer,
>
> Thanks for your valuable comments. We address each of your points below.
>
> ***Q1.** The local training, querying and retraining approach is taken as granted. However, it is not clear to me that this should be the default, or even a popular algorithm for so-called heterogeneous FL. The authors cited (Li \& Wang, 2019; Zhu et al., 2021), but the approaches in the two papers seem to be different from Figure 1 (without considering the encryption).*
>
> **A1.** We would like to clarify that our GuardHFL follows a general paradigm of heterogeneous FL (HFL), which is proposed by [Li \& Wang, 2019]. This paradigm contains three stages: local training, querying and local re-training.
> The only difference between GuardHFL and other HFL works lies in the acquisition of query samples in the querying stage.
> We give a comparison below, and also discuss it in detail in Section 2.1 of the revision.
>
> In general HFL, there is a large public auxiliary dataset (used as query samples) that every party can access.
> However, considering the privacy limitation, such public datasets are hard to collect in real-world scenarios such as healthcare.
> To tackle this problem, in GuardHFL, each party can locally construct a synthesized querying set based on his private training set, by utilizing existing data augmentation strategies (refer to Section 3.4 in the manuscript).
> In summary, GuardHFL is in line with the general HFL framework with the additional benefit of privacy protection.
> Moreover, although some variants of HFL have been proposed [1] [Zhu et al., 2021], they essentially still follow this three-stage paradigm described above.
>
> ***Q2.** I would encourage the authors to think about ways to justify the training paradigm, and make stronger connections between the training paradigm and the proposed method.*
>
> **A2.** Thanks for your comment.
> The main goal of this work is to design efficient and customized protocols for the training of heterogeneous FL.
> As explained in Answer 1, GuardHFL is built upon the standard HFL training paradigm, which contains three stages: local training, querying and local re-training.
> Since the local training and re-training stages are evaluated locally without revealing privacy, GuardHFL focuses on formalizing the querying stage of HFL.
> In generally, we generate each party's query samples using data augmentation strategies on his local training set, rather than relying on public datasets, and further design customized privacy-preserving protocols for this stage.
> We have revised the manuscript including Introduction and Section 3, and also provide a detailed analysis in the following.
>
> In detail, the querying party $P_Q$ first constructs querying samples based on his local training set (Section 3.4).
> Since querying samples imply the semantic information of private training data, they cannot be directly exposed to the server and the responding party $P_A$ for prediction.
> Therefore, GuardHFL secret-shares the query samples to the server and $P_A$ using the designed secure query-data sharing protocol (Section 3.1).
> Then given the secret-shared samples, $P_A$, $P_Q$ and the server can jointly execute the proposed secure model prediction protocol (Section 3.2) to obtain the secret-shared inference logits.
> After that, the secure result aggregation protocol (Section 3.3) comes in handy, which takes as input the secret-shared logits and returns the aggregated results to $P_Q$.
>
> During the entire querying phase, GuardHFL maintains the following invariant: the server and $P_A$ start each protocol with secret-shared inputs, and end with secret-shared outputs.
> This allows GuardHFL to sequentially stitch the proposed protocols to obtain a fully private querying scheme.
>
>
> [1] Ensemble Distillation for Robust Model Fusion in Federated Learning. [NeurIPS, 2020].

---

> ### Author Response · Authors · 2022-11-15
> **Response to Reviewer VwDD (2/2)**
>
> ***Q3.** I would also encourage the authors to discuss more about the application setting, and limitation of the proposed method.*
>
> **A3.** Thanks for your comment.
> We have added a discussion of our limitations in Section 5 in the revision, in terms of communication cost and malicious adversaries.
>
> (1) GuardHFL requires interactive communication between clients and the server, which is also an inherent limitation of HFL due to the query-response mechanism.
> As shown in Algorithm 1, during each iteration, each party needs to play the roles of the querying party to communicate with others within multiple rounds.
> Therefore, GuardHFL is more suitable for the cross-silo applications as stated by the reviewer.
> One possible mitigation is to extend the insight of the $k$-regular graph in FL [2] to HFL, and carefully design protocols from scratch.
> The main idea is that in FL it is enough for each party to speak to $k<n-1$ other parties via the server, where $n$ is the number of parties.
>
> (2) In line with many prior works [3] [4], GuardHFL considers an honest-but-curious adversary setting, where each entity (including the clients and the server) strictly follows the specification of the designed protocol but attempts to infer more knowledge about these private information of other clients.
> The security of GuardHFL may be improved to defeat more powerful malicious adversaries. Unfortunately, even using the best-known technique [5], the overhead will be increased by several orders of magnitude.
>
> The above discussions are interesting and we leave them as future work.
>
> [2] Secure Single-Server Aggregation with (Poly)Logarithmic Overhead. [CCS, 2020].
>
> [3] Federated model distillation with noise-free differential privacy. [IJCAI, 2021].
>
> [4] CaPC Learning: Confidential and Private Collaborative Learning. [ICLR, 2020].
>
> [5] Swift: Super-fast and robust privacy-preserving machine learning. [USENIX Security, 2021].

---

### Decision · Program_Chairs · 2023-01-20

**Decision:**

Reject

**Justification For Why Not Higher Score:**

I'm not enough of an expert and the only reviewer with sufficient relevant expertise did not like the work.

**Justification For Why Not Lower Score:**

N/A

**Metareview: Summary, Strengths And Weaknesses:**

This work considers so-called Heterogeneous Federated Learning which is a variant of Multi-Silo FL in which parties train different models on  their data and can query each other to improve their models. In this work a framework from (Li and Wang 2019) is used in which models are queried on public samples or samples generated from local data via augmentation. To preserve privacy in this scheme secret-sharing based cryptographic protocols are used to perform inference without revealing samples/predictions. This work proposes computational improvements to such a secret-sharing inference protocol.
The scope of the problem setting is rather narrow and the nature of these improvements, is primarily within the area of secure multi-party-computation. As a result none of the reviewers had both sufficient confidence as positive evaluation of this work. I therefore recommend rejection.